https://doi.org/10.1038/s41467-020-14941-6　　**OPEN**

# A genome-wide approach for identification and characterisation of metabolite-inducible systems

Erik K.R. Hanko [1], Ana C. Paiva [2], Magdalena Jonczyk [1], Matthew Abbott [1], Nigel P. Minton [1] & Naglis Malys [1✉]

Inducible gene expression systems are vital tools for the advancement of synthetic biology. Their application as genetically encoded biosensors has the potential to contribute to diagnostics and to revolutionise the field of microbial cell factory development. Currently, the number of compounds of biological interest by far exceeds the number of available biosensors. Here, we address this limitation by developing a generic genome-wide approach to identify transcription factor-based inducible gene expression systems. We construct and validate 15 functional biosensors, provide a characterisation workflow to facilitate forward engineering efforts, exemplify their broad-host-range applicability, and demonstrate their utility in enzyme screening. Previously uncharacterised interactions between sensors and compounds of biological relevance are identified by employing the largest reported library of metabolite-responsive biosensors in an automated high-throughput screen. With the rapidly growing genomic data these innovative capabilities offer a platform to vastly increase the number of biologically detectable molecules.

[1] BBSRC/EPSRC Synthetic Biology Research Centre (SBRC), School of Life Sciences, The University of Nottingham, Nottingham NG7 2RD, UK. [2] Centre for Biomolecular Sciences, School of Life Sciences, The University of Nottingham, Nottingham NG7 2RD, UK. ✉email: n.malys@gmail.com

Inducible gene expression systems execute a pivotal role in establishing a sustainable balance of gene expression and protein synthesis at the genome or single pathway/circuit level in response to changes in the intra- and extracellular environment. Such systems have been historically utilised for gene overexpression and protein production. Nowadays, inducible systems and their underlying genetic elements have become essential tools in synthetic biology[1]. Initially harnessed for the design of synthetic regulatory circuits[2], metabolite-responsive transcriptional regulators (TRs) and their cognate inducible promoters have received increasing attention due to their application as genetically encoded biosensors[3–6]. Although whole-cell biosensors or cell-free transcription/translation systems have advanced the fields of clinical diagnostics, environmental remediation, spatiotemporal regulation of signalling networks and metabolic engineering[7–11], the number of compounds that can be detected is still limited. Thus, to increase the diversity and to offer unique specificities, previously uncharacterised inducible systems must be sought and researched.

In instances, where an inducible system is to be found for a specific effector molecule, transcriptome analyses[12,13] and the evaluation of promoter libraries[14] have proven to be efficient strategies to discover effector-responsive promoters. They do not, however, exclude promoters that are indirectly activated nor guarantee identification of their associated TRs. Some of these issues may be solved by cloning sequence clusters containing TR-promoter pairs as demonstrated in the screening of metagenome libraries[15], but this methodology relies on the inducible system being functional in an organism different to the one it was sourced from. The reverse strategy relies on predicting the effector molecule based on genetic context[16] or comparative genomics[17,18]. This approach has successfully resulted in the identification of effectors and their corresponding TR-promoter pairs, but is limited to specific families of TRs and specific classes of compounds.

In this work, we address the deficiencies associated with the identification of metabolite-responsive-inducible systems by interconnecting information on ligand metabolism, TR genes and gene clusters responsible for the catabolism of the corresponding ligand. A generalised genome-wide approach is established to discover previously uncharacterised systems independent of their belonging to a specific family of regulators, the class of compounds they respond to or bacterial species utilised as a genetic resource. The discovered systems are validated for their response to proposed ligands and a comprehensive characterisation is performed. Specifically, we demonstrate their broad-host-range applicability in three industrially relevant microorganisms and address a typical issue that may arise, when employing a system in a host organism different to the one it was mined from. To facilitate forward engineering efforts, the identified inducible systems are parameterised and we demonstrate their utility for controlling orthogonal gene expression. We highlight their potential to be applied for investigation of metabolism and to expand the number of biologically detectable chemical species by evaluating the cross-reactivity between the library of biosensors and a comprehensive list of selected compounds. Finally, the biosensor responding to the industrially important intermediate compound β-alanine is applied to screen a library of L-aspartate 1-decarboxylase homologues and enzymes with superior activities are identified.

## Results

**A genome-wide approach to identify inducible systems.** Transcription factor-based-inducible systems are composed of a TR protein and an inducible promoter, including TR and RNA polymerase binding sequences. In the systems that control gene clusters associated with metabolism and catabolism in particular, the level of gene expression from the inducible promoter is often controlled by the TR that responds to small effector molecules, also referred to as ligands. To make such systems universally applicable, all three components need to be identified: the regulator, the inducible promoter and its corresponding effector.

For the identification of inducible systems, we chose the highly conserved genetic arrangement, typical of LysR-type TRs (LTTRs), but not exclusive to other types of TRs, to serve as a platform for the genome scale approach. In this commonly occurring arrangement, TRs are transcribed in divergent orientation of target genes or operons[19,20]. Once a complete list of annotated genes belonging to one species is retrieved from GenBank[21] (www.ncbi.nlm.nih.gov), including information on coding strand orientation and protein function, it is screened for TRs that are oriented in the opposite direction of operons involved in metabolism of any or specific ligands. To constrain the search, the operon itself is to be composed of at least two genes encoding annotated catalytic functions associated with a distinct metabolic pathway. For each enzyme encoded by the operon, a list of metabolic substrates and products is extracted from The Comprehensive Enzyme Information System[22] (BRENDA, www.brenda-enzymes.org). By comparing potential metabolite substrates and products of each of the involved enzymes, the primary substrate is concluded that is likely to be metabolised by the operon-encoded enzymes. This compound was proposed to be the ligand that binds the TR, initiating expression of genes that encode the ligand-metabolising pathway enzymes (Fig. 1). By following this approach, the TR is assigned a role in metabolism solely based on its proximity to a metabolic cluster of genes. To exemplify the utility of the approach, it was applied in the chemolithoautotrophic bacterium *Cupriavidus necator* H16, known for its metabolic versatility and diverse gene expression regulation. Consequently, 16 putative metabolite-responsive transcription factor-based inducible gene expression systems were identified (Fig. 2a, b; Supplementary Table 1). Their genomic organisation is illustrated in Supplementary Fig. 1.

**Validation of inducible systems and quantitative evaluation.** By the genome-wide analysis identified native systems were cloned into a modular reporter vector to examine their response to the presence of the proposed compounds. The original genetic organisation was conserved by positioning the TR-coding sequence in the opposite orientation of a reporter gene

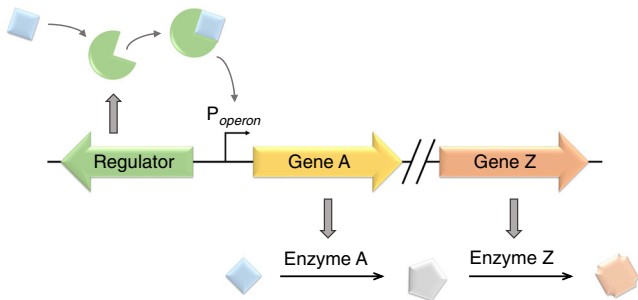

**Fig. 1 Schematic illustration of a TR controlling expression of a metabolic cluster of genes.** The primary substrate was proposed to be the ligand (light blue diamond). Note that the enzyme that converts the primary substrate into an intermediate product (grey regular pentagon) may be encoded by any of the genes in the operon. TR gene and protein are shown as green left arrow and pie, respectively. Metabolic cluster genes are yellow and orange right arrows. Product of metabolic conversion is shown as an orange octagon.

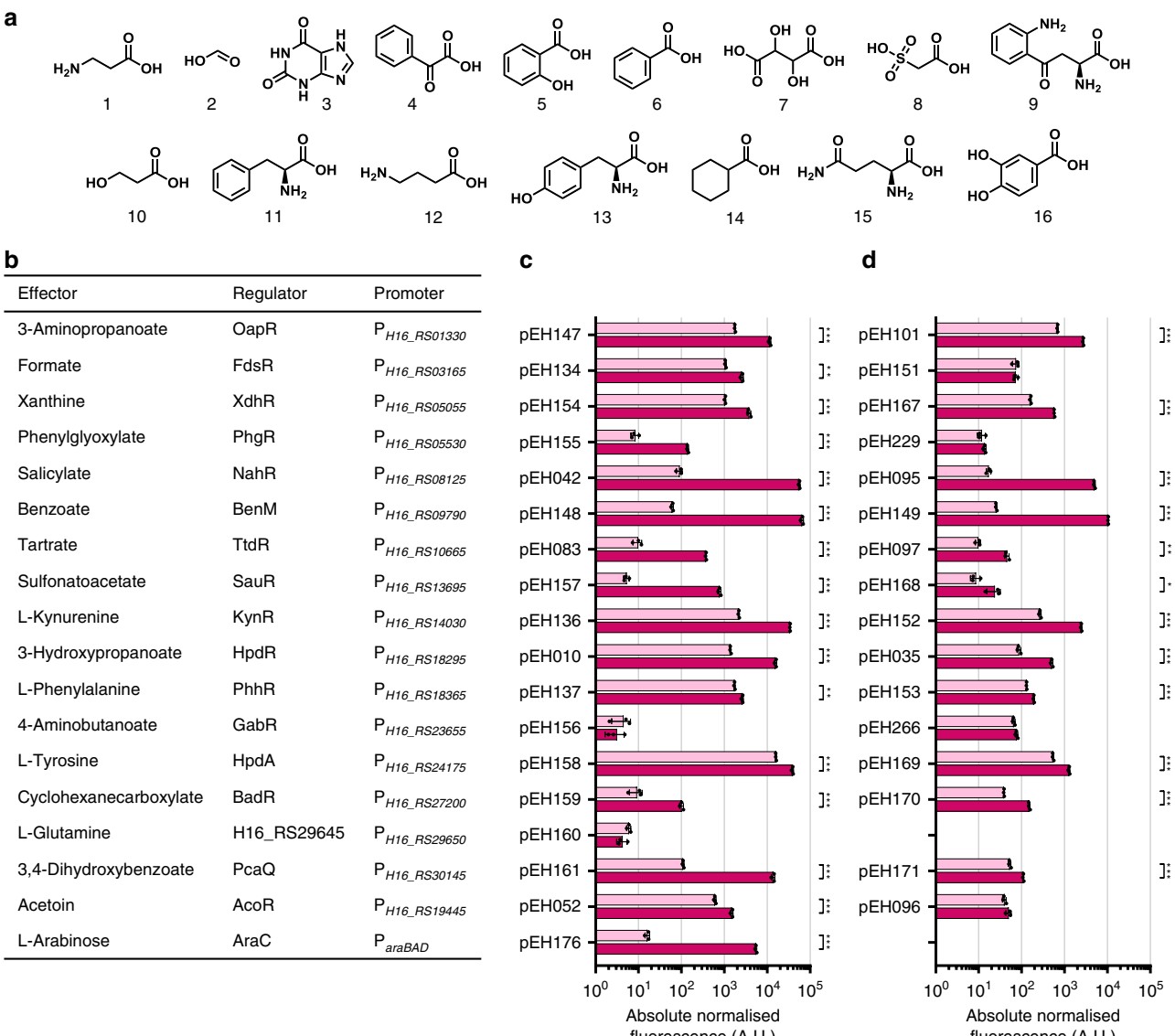

**Fig. 2 Quantitative evaluation of native inducible systems. a** Chemical structures of the proposed primary effector molecules: β-alanine (1), formic acid (2), xanthine (3), phenylglyoxylic acid (4), salicylic acid (5), benzoic acid (6), tartaric acid (7), sulfoacetic acid (8), L-kynurenine (9), 3-hydroxypropionic acid (10), L-phenylalanine (11), γ-aminobutyric acid (12), L-tyrosine (13), cyclohexanecarboxylic acid (14), L-glutamine (15) and 3,4-dihydroxybenzoic acid (16). **b** Summary of the identified inducible systems, including the inducible promoter, TR name and corresponding ligand. **c** Single time-point RFP fluorescence measurements (arbitrary units) of *C. necator* H16 carrying the transcription factor-based inducible gene expression systems composed of TR and inducible promoter in the same order as listed in **b**. The plasmids harbouring the individual system–reporter constructs are indicated. **d** Single time-point RFP fluorescence measurements of *C. necator* H16 carrying the inducible 'promoter only' implementations in the same order as listed in **b**. The plasmids harbouring the individual promoter–reporter gene constructs are indicated. Fluorescence output was determined in the absence of inducer (light magenta) and 6 h after extracellular supplementation with the corresponding effector to a final concentration of 5 mM (dark magenta). Data are mean ± SD, $n = 3$, $*p < 0.05$, $**p < 0.001$, $***p \leq 0.0001$, unpaired two-tailed $t$-test. Source data are provided as a Source Data file.

encoding a monomeric red fluorescent protein (mRFP)[23] (Supplementary Fig. 2). This arrangement enables reporter protein synthesis in response to exogenous supplementation with the proposed ligand to be measured by fluorescence output. The *Escherichia coli* L-arabinose-, and the *C. necator* acetoin-inducible systems have been tested previously[24,25] and were included for comparative purpose (Fig. 2b).

*C. necator* strains carrying the inducible systems were grown in rich medium and fluorescence output of the logarithmically growing cells was quantified 6 h after addition of the inducer. It should be noted that the metabolic gene cluster, which is putatively controlled by KynR, despite converting L-tryptophan into anthranilic acid and L-alanine, the intermediate compound

L-kynurenine was proposed to be the effector molecule and not the primary substrate L-tryptophan based on the thorough characterisation of KynR in other bacterial species[26]. Of the 16 analysed putative inducible systems, 14 showed a significant increase in mRFP protein synthesis after supplementation with their proposed effector molecules (Fig. 2c). Systems responding to 3-aminopropanoate (β-alanine) and phenylglyoxylate have not been reported, and highlight the potential of the developed approach for mining biosensors. Moreover, seven of the native systems exhibited higher levels of gene expression than the commonly used heterologous L-arabinose-inducible system in the presence of their corresponding effectors. Benzoate mediated the highest induction (factor of 1063-fold) and the highest absolute

normalised fluorescence with an expression level of >11-fold higher than AraC/$P_{araBAD}$. Low basal promoter activities were observed for metabolites that are neither involved in primary metabolism in *C. necator* nor likely to be present in the employed complex medium, including sulfonatoacetate, tartrate, cyclohexanecarboxylate and phenylglyoxylate.

However, the putative 4-aminobutanoate (GABA)- and L-glutamine-inducible systems showed no induction, even though their proposed ligands are involved in primary metabolism in *C. necator* and likely to be present in the rich medium. We hypothesised that translational start sites of the respective first gene in both operons are incorrectly annotated, resulting in reporter constructs with ineffectual 5′ untranslated regions. To test this hypothesis, the GABA- and L-glutamine-inducible systems were redesigned comprising the TR gene, the intergenic region, the first gene in the operon and the intergenic region preceding the second gene cloned upstream of the reporter (Supplementary Fig. 3a). Both inducible systems resulted in increased basal promoter activities (Supplementary Fig. 3b). Remarkably, GabR/$P_{H16\_RS23655}$ mediated a 1.9-fold induction of gene expression in the presence of GABA.

Increase in construct size can ultimately become the limiting factor in synthetic biology. As it has been reported, the transformation efficiency linearly decreases with increasing plasmid size[27]. Utilising an inducible system that is endogenous to the organism provides the advantage to possess a copy of the TR encoded in the genome enabling truncation of the controllable element to the sole inducible promoter, thus reducing construct size considerably. Despite that promoter activities under both uninduced and induced conditions generally decreased when the TR gene was removed from the multicopy episomal vector, a majority of inducible promoters significantly facilitated gene expression in the presence of their corresponding effectors (Fig. 2d). Whereas the 'promoter only' construct containing $P_{H16\_RS27200}$, which lacks the copy of the TR gene *badR*, exhibited a greater fluorescence level in the absence of effector than that of the BadR/$P_{H16\_RS27200}$-inducible system, suggests that BadR acts as a repressor, similarly to its homologue in *Rhodopseudomonas palustris*[28]. All other native TRs might be classed as activators or dual-function TRs. Furthermore, the induction factor was smaller for all of the systems, as a result of an altered ratio between transcription factor binding sites and available TR proteins. In fact, the promoters controlled by formate, phenylglyoxylate, GABA and acetoin showed no significant induction. Two of the inducible promoters, however, demonstrated an exceptionally strong activation of *rfp* expression: $P_{H16\_RS09790}$, responding to benzoate, and $P_{H16\_RS08125}$, which is activated by salicylate, mediated inductions by 403- and 292-fold, respectively. The 146 bp long intergenic region containing the benzoate-inducible promoter itself showed a stronger activation of gene expression than the majority of the 'complete' inducible systems, including the commonly used L-arabinose-inducible system (Fig. 2c). This characteristic highlights the potential of $P_{H16\_RS09790}$ to be employed as individual genetic element to control high levels of gene expression by reducing construct size by sevenfold. Even in cases, where a TR gene cannot be mapped to a cluster of genes involved in metabolism, the methodical approach described in this study can be employed for mining endogenous metabolite-inducible promoters.

**Demonstration of broad-host-range applicability.** To assess the potential of the constructed biosensors to be applied in other microorganisms, we evaluated the transcription factor-based-inducible systems that were mined from the genome of the β-proteobacterium *C. necator* in the industrially relevant γ-proteobacteria *E. coli* and *Pseudomonas putida*.

Regardless of their origin, the majority of systems responding to the 16 primary effectors, including acetoin, has never been tested in *E. coli* and *P. putida*. Thus, the broad-host-range applicability of the identified systems is highlighted by the outcome that a total of 8 and 12 of the 16 systems mediated a significant increase in reporter gene expression after inducer addition in *E. coli* (Fig. 3a) and *P. putida* (Fig. 3c), respectively. Three of them, activated by salicylate, benzoate and 3,4-dihydroxybenzoate induced by >75-fold in both tested microorganisms. Compared to systems that were sourced from other prokaryotes and tested in *E. coli* for controllable gene expression, PcaQ/$P_{H16\_RS30145}$ and BenM/$P_{H16\_RS09790}$ from *C. necator* outperform previously evaluated 3,4-dihydroxybenzoate- and benzoate-inducible systems by ~5- and 50-fold[15,29] (Supplementary Table 2). Specifically, benzoate mediated the highest induction (factor of 4428-fold) in *E. coli* and the highest absolute normalised fluorescence in *P. putida* with an expression level of >15-fold higher than AraC/$P_{araBAD}$ demonstrating its potential for high-level inducible gene expression across different species.

To test for regulator-dependant orthogonality the 'promoter only' versions of the inducible systems were evaluated for controllable gene expression in *E. coli* and *P. putida*. Without the episomal copy of the TR gene, none of the promoters showed an increase in activity even in the presence of the effector in *E. coli* (Fig. 3b), whereas in *P. putida* RFP synthesis was significantly induced from the phenylglyoxylate-, salicylate-, benzoate- and acetoin-controllable promoters (Fig. 3d). Activation of reporter gene expression may be explained by cross-reactivity of chromosomally encoded TRs. A protein blast revealed homologues of BenM and AcoR to be encoded in the genome of *P. putida* (Supplementary Table 3), which might be able to activate gene expression from the *C. necator* benzoate- and acetoin-inducible promoters, respectively. NahR and PhgR homologues could not be identified indicating that transcriptional activation from the salicylate- and phenylglyoxylate-inducible promoters may result from unspecific TR binding.

**Engineering the β-alanine-inducible system.** Each inducible system harbours several functional genetic elements that independently contribute to its overall performance. These regulatory elements, including promoters and ribosome binding sites (RBSs), which control the expression of the TR gene and its associated regulon, may vary in their usage and efficiency across different species. As a consequence, inducible systems may perform differently when transferred from one into another organism. For example, the β-alanine-inducible system mediated a moderate activation of reporter gene expression in *C. necator*, whereas in *E. coli* and *P. putida* the fluorescence output remained at basal levels even in the presence of the inducer (Fig. 3a, c). β-Alanine is an intermediate compound for the synthesis of industrially relevant nitrogen-containing platform chemicals, including acrylamide, acrylonitrile and poly-β-alanine (also known as nylon-3)[30,31]. Furthermore, it is a precursor of the dipeptides carnosine and anserine that have been demonstrated to improve cognitive functions and physical capacities in humans[32,33]. To expand the host range and due to its usefulness as biosensor in synthetic biology and biotechnology applications, the β-alanine-inducible system from *C. necator* was pursued to be modified to enable its utilisation for gene expression control in *E. coli* and *P. putida*.

Although the system did not demonstrate an increase in RFP synthesis after addition of β-alanine, the promoter $P_{H16\_RS01330}$ showed the fifth highest activity of all evaluated systems under

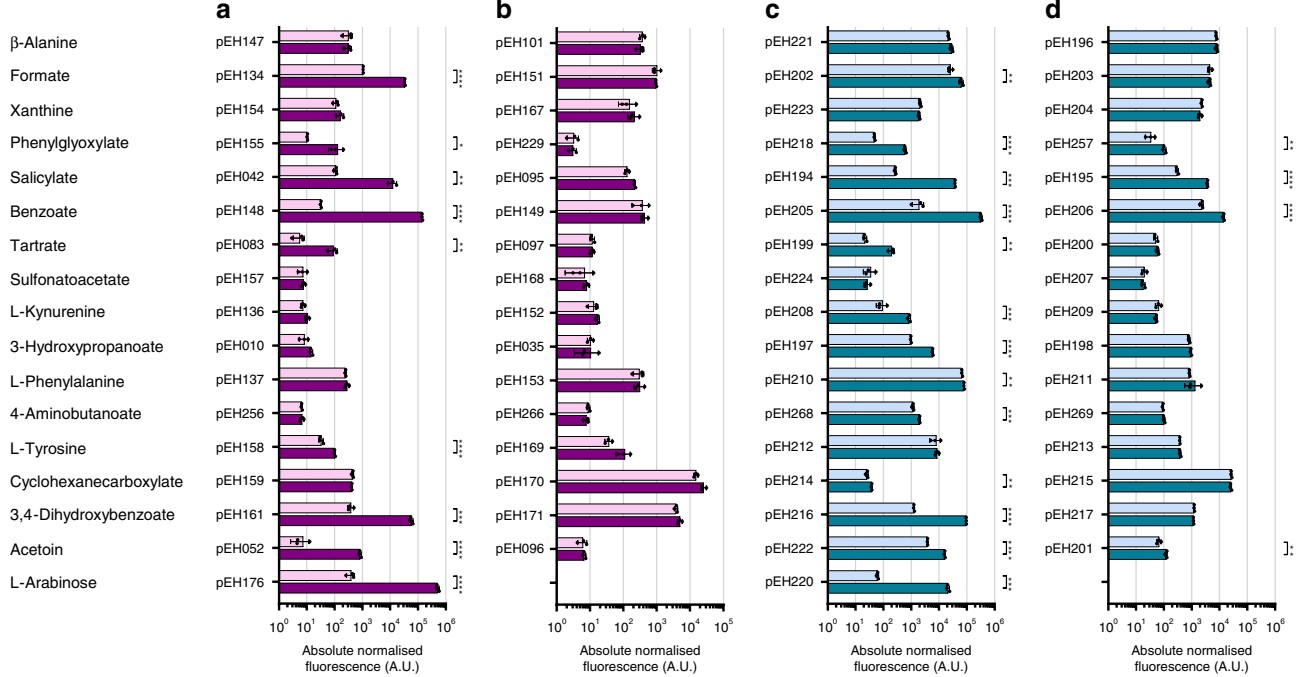

**Fig. 3 C. necator native systems mediate controllable gene expression in E. coli and P. putida.** Single time-point fluorescence measurements of **a**, **b** E. coli and **c**, **d** P. putida carrying the C. necator-inducible gene expression systems composed of TR and promoter **a**, **c** or 'promoter only' version **b**, **d**. The plasmids harbouring the individual system- or promoter–reporter gene constructs are indicated. RFP fluorescence output was determined in the absence of inducer (light purple and light blue) and 6 h after extracellular supplementation with the corresponding primary effector to a final concentration of 5 mM (dark purple and dark blue). Data are mean ± SD, n = 3, *p < 0.05, **p < 0.01, ***p ≤ 0.001, ****p ≤ 0.0001, unpaired two-tailed t-test. Source data are provided as a Source Data file.

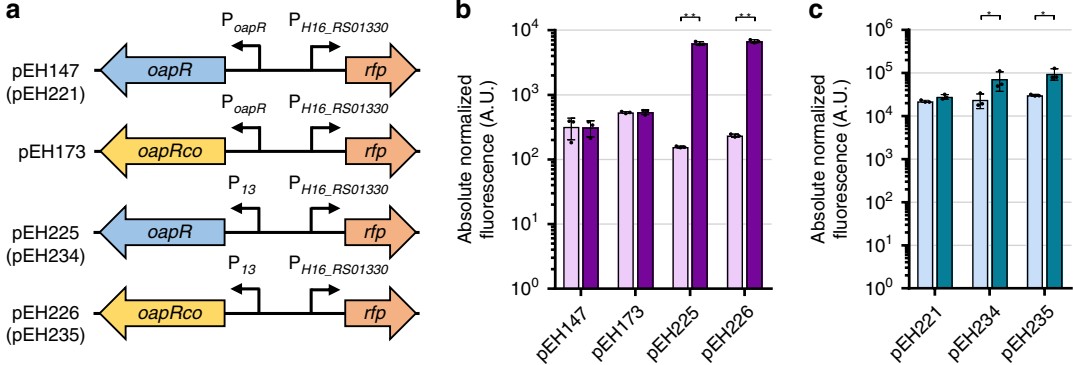

**Fig. 4 Engineering the β-alanine-inducible system. a** Schematic illustration of the different versions of the β-alanine-inducible system and their corresponding plasmid identifiers. Absolute normalised fluorescence of **b** E. coli and **c** P. putida carrying different versions of the β-alanine-inducible system–reporter construct in the absence (light purple and light blue) and presence (dark purple and dark blue) of 5 mM β-alanine. Single time-point RFP fluorescence measurements were taken 6 h after effector addition. Data are mean ± SD, n = 3, *p < 0.05, **p < 0.0001, unpaired two-tailed t-test. Source data are provided as a Source Data file.

uninduced conditions in E. coli (Fig. 3a). This indicated that the regulatory elements of $P_{H16\_RS01330}$ are functional in E. coli and that the lack of induction is more likely to be attributed to inducer uptake or regulator gene expression. A poor ligand transport as a cause of the absence of induction could be excluded, as β-alanine has been reported to be actively taken up by the cells[34]. Strikingly, the GC-content of the TR-coding sequence (71%) is significantly higher than the GC-content of the E. coli K12 genome (51%). To rule out that a high GC-content impairs TR synthesis, H16_RS01325 (oapR) was codon optimised for E. coli codon usage (pEH173, Fig. 4b). However, this modification did not improve the response of the system to β-alanine (Fig. 4b).

Subsequently, to ensure that the regulator is expressed in E. coli and P. putida, the C. necator native promoter of the TR was replaced by a host-specific promoter. The 24 bp DNA sequence upstream of the oapR translational start site was replaced by the core sequence of a medium-strength insulated constitutive promoter, $P_{13}$ (ref. [35]), including the phage T7 gene 10 RBS. The substitution was implemented in both plasmids containing the native (pEH147 and pEH221) and the codon-optimised oapR (pEH173, Fig. 4a). The addition of β-alanine resulted in a 40- and 29-fold increase in fluorescence output for E. coli cultures carrying pEH225 and pEH226, respectively (Fig. 4b). This suggests that the original promoter $CnP_{oapR}$ is not functional in E. coli and that codon optimisation might even lead to a lower TR

synthesis rate, which results in a decreased induction level. Moreover, a lower background reporter gene expression under uninduced conditions in case of pEH225 and pEH226 indicates that OapR, a member of the MocR family, may act as dual-function TR, repressing transcription of *oapTD* in the absence of β-alanine, but acting as activator in its presence. The dual mode of action has been observed for other members of this family of TRs[36]. In *P. putida*, the promoter and RBS exchange also resulted in a threefold induction of gene expression for both versions the native (pEH234) and the codon-optimised (pEH235) *oapR* (Fig. 4a, c). In contrast to *E. coli*, however, induction levels are significantly lower that may be attributed to a higher basal promoter activity in *P. putida*.

**Parameterisation of inducible systems**. The 16 functional native inducible systems, including the acetoin-inducible system, were subsequently evaluated for their dose–response, dynamic range, induction homogeneity and orthogonality. To obtain a preliminary overview of the kinetics of induction and to assess the effect of the extracellularly added ligand on cell growth, *C. necator* strains carrying plasmids with inducible systems were grown in minimal medium (MM) supplemented with the corresponding inducer at a final concentration of 5 mM, and fluorescence and absorbance were monitored over time. For all inducible systems, increasing fluorescence above the level of the uninduced culture was observed within 30 min after the effector had been added (Supplementary Fig. 4). Maximum fluorescence was reached within 2–4 h for most of the systems with exception of the sulfonatoacetate-, L-phenylalanine-, and cyclohexanecarboxylate-inducible systems continuing to produce the reporter protein after 6 h. Most of the compounds had a beneficial effect on growth and no toxicity was observed for any effector at the tested concentration of 5 mM (Supplementary Fig. 5). L-tyrosine and 3,4-dihydroxybenzoate had the most significant impact on growth resulting in more than a twofold increase in cell density most likely due to ligand catabolism. Therefore, the effector consumption plays an important role in the kinetics of induction. In order to parameterise the identified systems, assumptions must be implemented that account for these factors, including ligand uptake and metabolism.

The dose–response curve of a metabolite-responsive-inducible system describes the level of gene expression as a function of ligand concentration, thus indicating the range of effector concentration in which the inducible system is able to operate. It provides key parameters that aid in part selection and the computational design of synthetic circuits. To simplify mathematical modelling approaches, effector concentrations are usually considered constant, mediating a sustained gene expression throughout the course of cell growth. In this study, however, the ligands are metabolised by *C. necator* resulting in a decrease in gene expression to basal levels once the inducer has been depleted. Therefore, the time point, at which the reporter output is correlated with the ligand concentration, must be chosen carefully. By postulating that the inducer-metabolising enzymes are not synthesised faster than the primary induction of reporter gene expression, and to account for the minimum amount of time required for RFP synthesis and maturation, the minimal induction interval of 80 min was determined. *C. necator* strains harbouring the inducible systems were grown in MM and reporter gene expression was monitored after supplementation with the corresponding inducer at a wide range of concentrations over time. The dose–responses were obtained by plotting the relative normalised fluorescence values of the 80-min minimal induction interval as a function of inducer concentration (Fig. 5a). Data points were fit using a Hill function (see Methods section),

taking into account the basal level of fluorescence output of the uninduced cells. Consequently, on the basis of the mathematically modelled dose–response curve, key parameters that distinguish one inducible system from another were obtained (Table 1).

One of the most important parameters when choosing an inducible system to tightly control different levels of gene expression is the dynamic range. It is defined as the maximum level of reporter output relative to basal expression levels (see Methods section, formula (3)). We found that the salicylate-inducible system has the highest dynamic range of the evaluated native systems followed by SauR/$P_{H16\_RS13695}$ and TtdR/$P_{H16\_RS10665}$. In general, the dynamic range is higher than the induction level of cells grown in rich medium at an effector concentration of 5 mM (Fig. 2c). This effect does not apply to the benzoate- and 3,4-dihydroxybenzoate-inducible systems, which show a lower dynamic range in MM. Whereas a lower induction level in complex medium might be the consequence of structurally similar molecules that are able to interact with the respective TR, the lower dynamic range in MM might be attributed to catabolic repression as it has been demonstrated to be the case for both benzoate and 3,4-dihydroxybenzoate in *P. putida*[37]. In addition to a high dynamic range in MM, NahR/$P_{H16\_RS08125}$ has the lowest $K_m$ of all evaluated native inducible systems. This parameter is defined as the inducer concentration that mediates half-maximal reporter output, suggesting that only small quantities of salicylate are needed to induce the system. Similar effector concentrations have been shown to mediate gene expression from the *P. putida* salicylate-inducible system NahR/$P_{sal}$[38]. In contrast to most of the inducible systems that operate in the μM-range, GabR, TtdR and SauR seem to respond to effector concentrations three to five orders of magnitude higher than NahR. Moreover, in case of GabR, the $K_m$ appeared to be higher than the concentration of GABA the growth medium can be supplemented with. It should be noted that the extracellular effector concentration may not necessarily correlate with the ligand concentration inside the cell ultimately dictating the level of gene expression. Ligand uptake limitations may therefore result in inaccurate parameters, as it might be the case for the GABA-, tartrate- and sulfonatoacetate-inducible systems. For the other 13 systems ligand uptake is assumed not to be limiting. The Hill coefficient *h* indicates the range of inducer concentration over which the system results in a change in reporter output. Inducible systems with a low Hill coefficient, such as SauR/$P_{H16\_RS13695}$, PhhR/$P_{H16\_RS18365}$ or HpdR/$P_{H16\_RS18295}$ exhibit a flatter dose–response function, indicating that gene expression is tuneable over a wider range of inducer concentration. On the contrary, systems with a higher Hill coefficient, including BenM/$P_{H16\_RS09790}$ and AcoR/$P_{H16\_RS19445}$, show a steeper dose–response function suggesting that they behave more like an on/off switch.

The homogeneity of induction was evaluated by flow cytometry. This method allows to determine whether intermediate inducer concentrations give rise to subpopulations of uninduced and fully induced cells. Their existence may indicate a more complex type of transcriptional regulation or inducer transport limitations[39]. Cell cultures of *C. necator* carrying the 16 functional native system–reporter constructs were subjected to inducer concentrations corresponding to 50 and 95% of the maximum level of fluorescence output $b_{max}$ (except in case of the phenylglyoxylate-, tartrate-, sulfonatoacetate- and GABA-inducible systems; refer to the legend of Fig. 5b; Supplementary Fig. 6). The reporter output was measured 2 h after the inducer had been extracellularly added. Each of the 16 evaluated systems demonstrated a unimodal induction behaviour after addition of the corresponding ligand at both medium and nearly saturating concentrations. Generally, the fluorescence

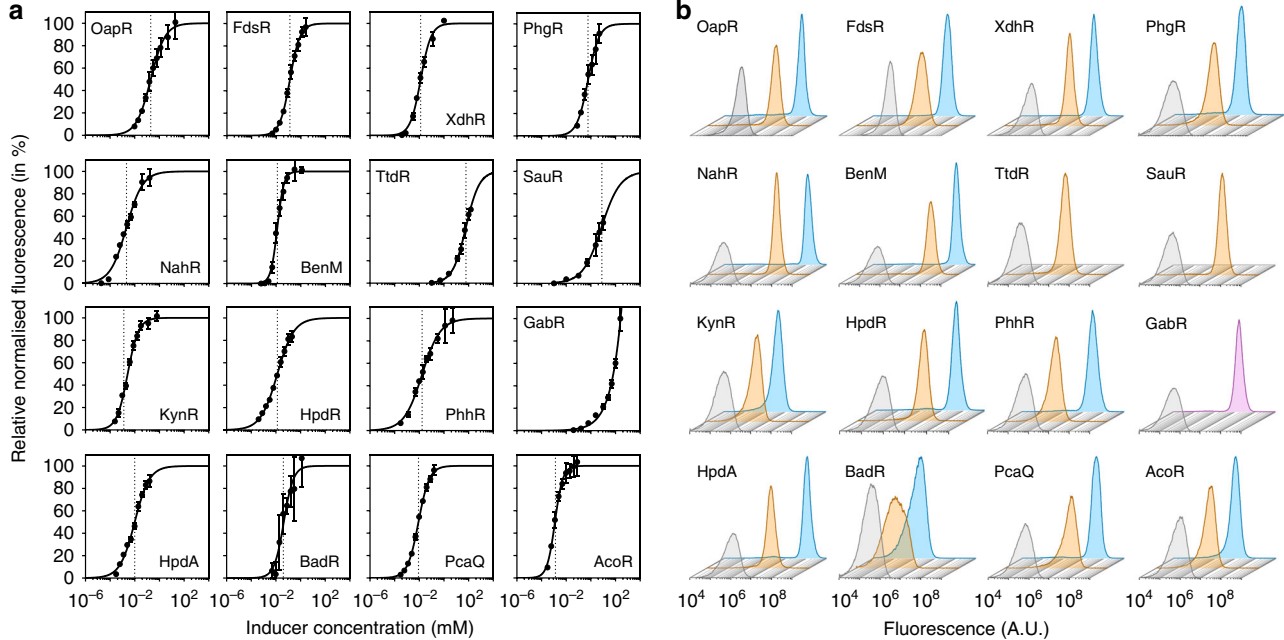

**Fig. 5 Response function and induction homogeneity. a** Relative normalised fluorescence of *C. necator* carrying the various inducible system–reporter constructs in response to different concentrations of their corresponding primary inducers. Measurements were taken 80 min after the inducer had been extracellularly added. The dose–responses were fit using a Hill function (see Methods section). The maximum level of reporter output $b_{max}$ was set to 100% (except in case of the GABA-inducible system where the absolute normalised fluorescence corresponding to the highest GABA concentration tested was set to 100%). The inducer concentration that mediates half-maximal reporter output $K_m$ is indicated by a dotted line. Data are mean ± SD, $n = 3$, non-linear regression. Source data are provided as a Source Data file. **b** Evaluation of induction homogeneity by flow cytometry. The fluorescence intensity of 100,000 individual cells was determined for each inducible system–reporter construct 2 h after extracellular addition of inducer. Uninduced cells (grey) are compared to cultures supplemented with their cognate effector at final concentrations corresponding to 50% (orange) and 95% (blue; 90% in case of the phenylglyoxylate-inducible system) of the maximum level of reporter output $b_{max}$. The tartrate- and sulfonatoacetate-inducible systems were only evaluated for induction homogeneity at 50% of $b_{max}$ due to solubility limits and inducer toxicity, respectively. Since $b_{max}$ could not be calculated for the GABA-inducible system, induction homogeneity was determined using a final concentration of 250 mM (purple).

### Table 1 Parameters of the native inducible systems.

| Inducible system | Inducer | Dynamic range, in -fold[a] | $K_m$[b] | $h$[c] |
|---|---|---|---|---|
| OapR/P$_{H16\_RS01330}$ | β-Alanine | 8.0 ± 0.4 | 201 ± 24 μM | 0.75 ± 0.05 |
| FdsR/P$_{H16\_RS03165}$ | Formate | 4.5 ± 0.2 | 130 ± 7 μM | 1.05 ± 0.04 |
| XdhR/P$_{H16\_RS05055}$ | Xanthine | 16.0 ± 1.0 | 13.0 ± 1.4 μM | 1.03 ± 0.10 |
| PhgR/P$_{H16\_RS05530}$ | Phenylglyoxylate | 144.8 ± 106.5 | 595 ± 133 μM | 0.96 ± 0.13 |
| NahR/P$_{H16\_RS08125}$ | Salicylate | 650.8 ± 317.4 | 2.12 ± 0.49 μM | 0.66 ± 0.08 |
| BenM/P$_{H16\_RS09790}$ | Benzoate | 74.1 ± 10.7 | 12.6 ± 0.8 μM | 1.64 ± 0.14 |
| TtdR/P$_{H16\_RS10665}$ | Tartrate | 370.6 ± 232.2 | 61.5 ± 22.9 mM | 0.80 ± 0.09 |
| SauR/P$_{H16\_RS13695}$ | Sulfonatoacetate | 395.4 ± 193.7 | 7.3 ± 4.9 mM | 0.57 ± 0.06 |
| KynR/P$_{H16\_RS14030}$ | L-kynurenine | 26.9 ± 2.1 | 2.64 ± 0.18 μM | 0.98 ± 0.05 |
| HpdR/P$_{H16\_RS18295}$ | 3-Hydroxypropanoate | 25.2 ± 1.7 | 13.2 ± 1.9 μM | 0.64 ± 0.03 |
| PhhR/P$_{H16\_RS18365}$ | L-phenylalanine | 9.0 ± 0.8 | 16.9 ± 2.2 μM | 0.59 ± 0.04 |
| GabR/P$_{H16\_RS23655}$ | GABA | ND | ND | ND |
| HpdA/P$_{H16\_RS24175}$ | L-tyrosine | 38.3 ± 5.0 | 9.9 ± 2.2 μM | 0.72 ± 0.07 |
| BadR/P$_{H16\_RS27200}$ | Cyclohexanecarboxylate | 11.4 ± 10.9 | 43.8 ± 13.0 μM | 1.11 ± 0.28 |
| PcaQ/P$_{H16\_RS30145}$ | 3,4-Dihydroxybenzoate | 77.2 ± 29.8 | 8.63 ± 0.40 μM | 0.98 ± 0.03 |
| AcoR/P$_{H16\_RS19445}$ | Acetoin | 11.1 ± 1.6 | 1.23 ± 0.07 μM | 1.36 ± 0.10 |

Data are mean ± SD, $n = 3$. Source data are provided as a Source Data file. ND - not determined.
[a]Dynamic range is defined as the -fold increase in fluorescence calculated by dividing the maximum level of fluorescence output by the basal level of fluorescence output.
[b]$K_m$ represents the inducer concentration at which the half-maximal activation of the inducible system is achieved.
[c]$h$—Hill coefficient.

distribution of the uninduced cells is wider than the distribution in the presence of effector. This generalisation does not apply to the cyclohexanecarboxylate-inducible system that may be a further indicator of it being a repressor-based type of inducible system.

**Orthogonality of inducible systems**. To establish whether any of these systems can be used in combination to independently control expression of more than a single gene the activity of the ligands against non-cognate promoters was evaluated. A total of 21 inducible systems was selected to test for orthogonality in

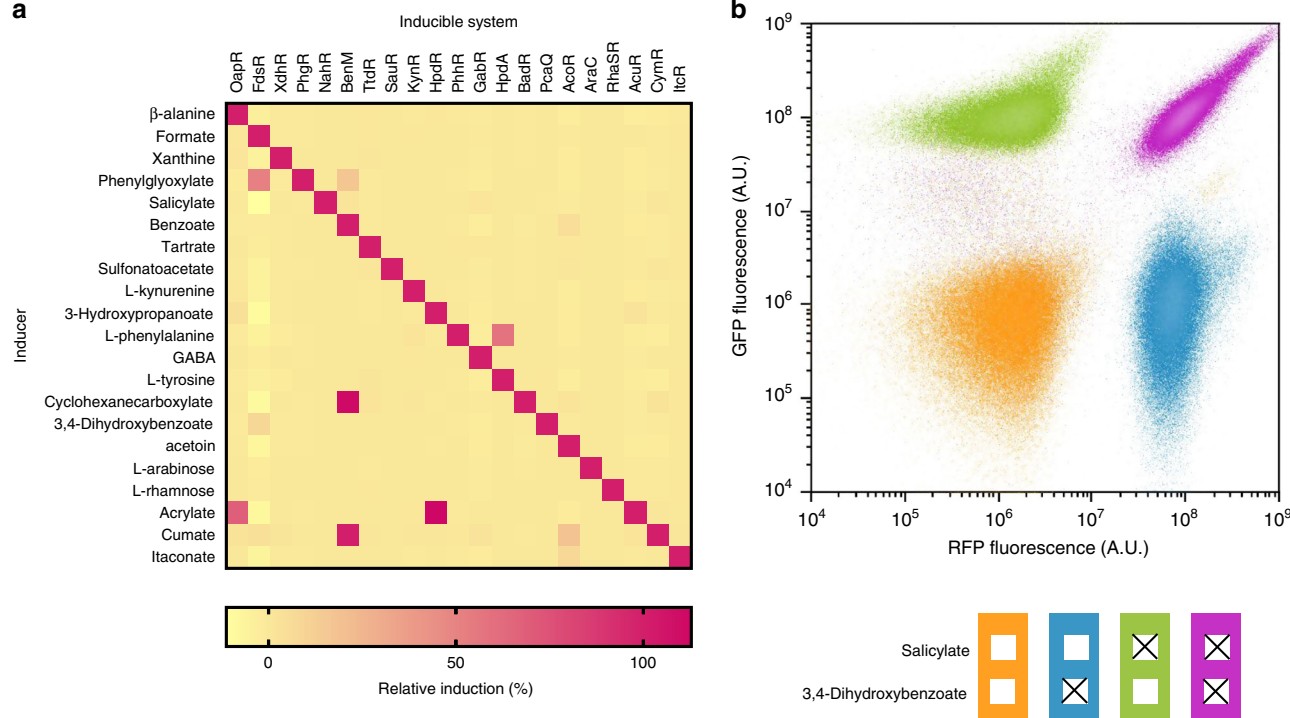

**Fig. 6 Orthogonality of inducible systems. a** Cross-reactivity of a set of 21 inducible systems and their corresponding primary inducers. The heat map illustrates induction of reporter gene expression in the presence of the metabolite (in %) relative to the induction mediated by the corresponding primary effector. Measurements were taken 6 h after supplementation with inducer at a final concentration of 5 mM. Data are mean, $n = 4$. Source data are provided as a Source Data file. **b** Fluorescence output of individual cells of *C. necator* carrying pEH263 measured by flow cytometry. The vector pEH263 contains the 3,4-dihydroxybenzoate- and salicylate-inducible systems controlling expression of *rfp* and *egfp*, respectively. Fluorescence was determined in the absence of inducer (orange), in the presence of 3,4-dihydroxybenzoate (blue), salicylate (green), and both 3,4-dihydroxybenzoate and salicylate (purple). Effector concentrations correspond to their respective $K_m$'s and fluorescence was measured 2 h after inducer addition.

*C. necator*. It includes the 16 native and the previously characterised heterologous L-arabinose-, L-rhamnose-, acrylate-, cumate- and itaconate-inducible systems[35,40].

To screen the 441 combinations of inducer and biosensor for cross-reactivity, we employed an automated platform (see Methods section). Strains of *C. necator* harbouring the inducible systems were grown in MM and transferred to a 96-well microtiter plate format. After the inducers had been extracellularly added, cells were cultured for 6 h before RFP fluorescence and cell density were measured. A total of 15 of the 21 inducible systems exhibited a strong affinity to their primary ligands, showing <5% cross-reactivity of non-target metabolites relative to the fluorescence output mediated by the primary effector (Fig. 6a, Supplementary Table 4). The remaining six inducible systems were activated by one or more metabolites other than their cognate inducers. This cross-reactivity may be the result of either structural resemblance, a metabolic relationship or a combination thereof. For example, L-phenylalanine is converted into L-tyrosine by the phenylalanine 4-monooxygenase PhhA (H16_RS18365). Therefore, activation of the L-tyrosine-inducible system by L-phenylalanine is more likely to be due to biological conversion of the added compound into the primary effector rather than direct interaction of L-phenylalanine with HpdA. However, since the difference between the two molecules lies in a singly hydroxyl-group, an induction by structural resemblance cannot be entirely ruled out. The same applies to phenylglyoxylate that activated the benzoate-inducible system. Phenylglyoxylate is converted by *C. necator* into benzoate via a two-step reaction with benzaldehyde as intermediate compound. During the first reaction, carbon dioxide is generated that may also explain induction of the

formate-inducible system by the structurally dissimilar phenylglyoxylate. Since carbon dioxide can subsequently be converted into formate[41], it is rational to postulate that the FdsR/P$_{H16\_RS03165}$-inducible system is activated in this case by its primary inducer. In addition to phenylglyoxylate, the benzoate-inducible system was activated by extracellular supplementation with cyclohexanecarboxylate and cumate. Cyclohexanecarboxylate shares both a structural resemblance to benzoate and downstream degradation pathways in *C. necator*, which makes it difficult to conclude its cause of cross-reactivity. In cases like these, we can take advantage of the system's transferability. Moving the inducible system from one organism into another host with a dissimilar metabolism may allow to distinguish more easily between induction by structural resemblance and metabolic relationship. To investigate the cause of cross-reactivity of the compounds that resulted in a fluorescence output of >10% relative to the primary inducer in *C. necator*, their induction behaviour was evaluated in *E. coli*. Single time-point fluorescence measurements of *E. coli* revealed that the benzoate-inducible system is activated by addition of cyclohexanecarboxylate and cumate (Supplementary Table 4). *E. coli* has not been reported to metabolise any of these compounds suggesting that cross-reactivity is caused by structural resemblance.

More difficult to explain is the cause of activation of the β-alanine- and 3-hydroxypropanoate (HP)-inducible systems by acrylate. It has been shown that *C. necator* is able to degrade acrylate[35]; however, relatively little is known about its metabolism. Activation by structural resemblance to the primary effector is less likely as addition of acrylate to *E. coli* carrying the engineered β-alanine-inducible system (pEH225) only resulted in

a relative induction of 2.7% in contrast to 68% in *C. necator* (Supplementary Table 4). Therefore, activation by structural resemblance of a degradation product or direct conversion into the primary effectors 3-HP and β-alanine in *C. necator* may be more likely. Acrylate can be activated into acryloyl-CoA by acyl CoA:acetate/3-ketoacid CoA transferase (H16_RS22005/H16_RS22010) or by the propionate CoA transferase Pct (H16_RS13535)[42], which, as it has been proposed by Peplinski et al.[43], can be converted into 3-HP via its CoA intermediate. However, pathways from acrylate or 3-HP to β-alanine or a metabolic intermediate, which is able to activate the β-alanine-inducible system, will have to be elucidated.

Based on the results of the cross-reactivity screen, two orthogonal inducible systems were employed to independently control expression of two fluorescent protein reporter genes. Plasmid pEH263 was constructed containing *rfp* under control of the 3,4-dihydroxybenzoate-, and *egfp* under control of the salicylate-inducible system (Supplementary Fig. 7). Cultures of *C. necator* carrying pEH263 were left uninduced, subjected to the individual inducers or the combination of both at final concentrations corresponding to $K_m$ (Table 1). The output of the two non-overlapping fluorescent proteins was determined by flow cytometry 2 h after the inducer/inducers had been extracellularly added. Employing every possible inducer combination, four distinct cell states could be observed (Fig. 6b, Supplementary Fig. 8). RFP and enhanced green fluorescent protein (eGFP) fluorescence in the absence of both inducers remained at background levels comparable to the single-system implementations (orange population). Addition of 3,4-dihydroxybenzoate resulted in synthesis of RFP only as represented by the blue population. Similarly, the presence of salicylate activated expression of *egfp* but not *rfp* (green population). The final cell state is represented by the purple population, where both inducers were added to mediate the simultaneous expression of both fluorescent protein reporter genes.

In conclusion, the identified native systems can be used in combination to independently control expression of multiple genes expanding the list of available switches in synthetic circuit design. Importantly, inducible systems for structurally similar (phenylglyoxylate, salicylate, cyclohexanecarboxylate and 3,4-dihydroxybenzoate) and distinctive (xanthine, tartrate, sulfonatoacetate and GABA) compounds were demonstrated to be fully orthogonal.

**Screening structurally and metabolically related compounds**. TRs often recognise molecules that are structurally similar to their primary effectors. For example, isopropyl β-D-1-thiogalactopyranoside is a commonly used structural analogue of allolactose, employed to control the expression of genes regulated by LacI. Here, the TR's specificity was investigated to identify previously uncharacterised ligand–TR interactions and consequently extend the biosensor application to the detection of structurally similar or metabolically related compounds. To do so, the library of 21 native and heterologous inducible systems was screened against 46 compounds using an integrated robotic platform, as described in the previous section. Of the 46 metabolites, 12 demonstrated an induction of at least 5% relative to the induction mediated by the corresponding primary effector and an absolute induction factor of at least five (Fig. 7, Supplementary Table 5). Similarly to the former orthogonality screen (Fig. 6a), it should be noted that activation of reporter gene expression by the extracellularly added compounds likely indicates a structural resemblance to the primary ligand, a metabolic relationship or a combination thereof. To shed light on their cause of system activation, cultures of *E. coli*

carrying the functional formate-, salicylate-, benzoate-, 3,4-dihydroxybenzoate-, acetoin- and β-alanine-inducible systems, active in this host organism, were evaluated for the same cross-reactivity (Supplementary Table 5).

Based on the results of this screen, 11 out of 12 metabolites may be classed into four groups. The first group comprises the compound/regulator pairs 3-aminobutanoate/OapR, DL-3-amino-2-hydroxypropanoate/OapR and cyclohexenecarboxylate/BenM. Neither of the three compounds has been reported to be metabolised by *E. coli* suggesting that they directly act as ligands. The response of the OapR/$P_{H16\_RS01330}$-inducible system to non-natural compounds, such as 3-aminobutanote and 3-amino-2-hydroxypropanoate, strongly supports our initial claim of the structurally nearly identical β-alanine (3-aminopropanoate) being the primary ligand of OapR. The other three groups contain the compound/regulator pairs that did not result in an induction in *E. coli*. For example, nicotinate, N-benzoylglycine, L-tryptophan and 4-hydroxybenzoate fall into the category of metabolites that are likely to be converted into the primary effectors in *C. necator* mediating gene expression from the formate-, benzoate-, L-kynurenine- and 3,4-dihydroxybenzoate-inducible system, respectively. None of these catabolic pathways exist in *E. coli*, which may explain their lack of induction. In the case of the hypoxanthine inducing the XdhR/$P_{H16\_RS05055}$-controllable system, it is less clear whether hypoxanthine itself interacts with XdhR, due to structural resemblance to xanthine, or if activation of gene expression is mediated by its catabolic product, the primary effector. In *Streptomyces coelicolor*, hypoxanthine was not able to bind the regulator of the gene cluster encoding xanthine dehydrogenase enzyme[44]. However, in contrast to the *S. coelicolor* XdhR, which belongs to the TetR family of TRs[45], the *C. necator* XdhR is a LTTR that may operate in a different manner. The last group comprises the remaining compounds that could not be confirmed in *E. coli*, but are likely to induce because of structural resemblance to the primary ligand. Direct interaction of cyclohexenecarboxylate and cyclopentanecarboxylate with BadR could not be confirmed due to dysfunctionality of BadR/$P_{H16\_RS27200}$ in *E. coli*. The other two compounds, including 2,6-dihydroxybenzoate and catechol, might not be taken up by, or diffused into, *E. coli* cells and the activation of reporter gene expression by metabolically related compounds may be excluded as their consecutive degradation products, resorcinol and *cis,cis*-muconate, respectively, were not able to induce the salicylate- and benzoate-inducible systems in *C. necator* (Fig. 7).

Lastly, significant activation of the acetoin-inducible system by L-lactate was observed in *C. necator*, but not in *E. coli*. Since there is no characterised metabolic pathway for L-lactate to be converted into acetoin, and the AcoR/$P_{H16\_RS19445}$-inducible system is highly activated by acetoin in both bacterial species, the response to L-lactate cannot be explained by either structural similarity or metabolic association.

**Biosensor-assisted screening of enzyme variants**. The β-alanine biosensor was employed to screen a library of six L-aspartate 1-decarboxylase (PanD) homologues for their ability to convert L-aspartate into β-alanine. The whole-cell enzymatic conversion of L-aspartate into β-alanine has been previously demonstrated in *E. coli*[46]. In this study, *C. necator* was selected as biocatalyst as it may allow for the autotrophic biosynthesis of this industrially relevant intermediate compound. To comparatively screen homologues of PanD, plasmids were constructed that contain the original β-alanine biosensor and each a *panD* gene under control of the L-arabinose-inducible promoter (Supplementary Fig. 9).

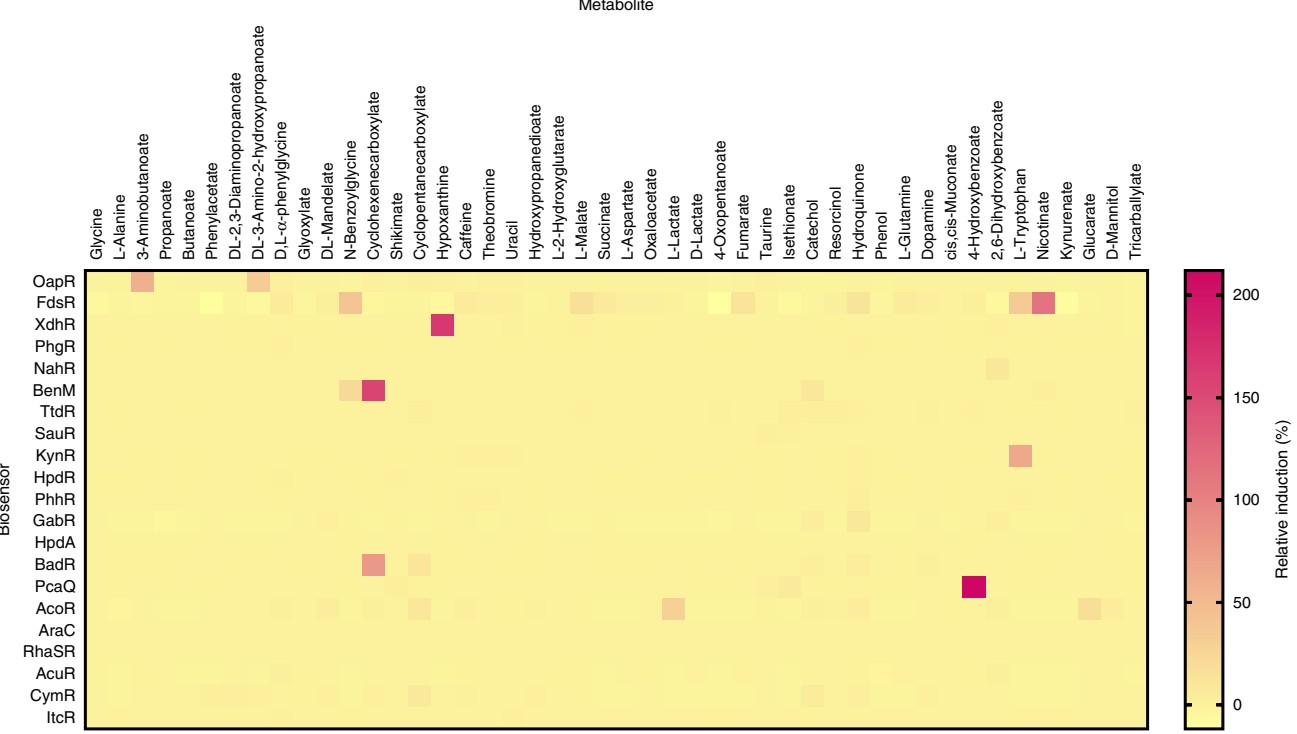

**Fig. 7 Screening of structurally similar and metabolically related compounds.** Cross-reactivity of a set of 21 inducible systems and a library of 46 compounds. The heat map illustrates induction of reporter gene expression in the presence of metabolite (in %) relative to the induction mediated by the corresponding primary effector. Measurements were taken 6 h after supplementation with inducer at a final concentration of 5 mM (except in case of L-2-hydroxyglutarate that was added at a final concentration of 0.5 mM). Data are mean, $n = 4$. Source data are provided as a Source Data file.

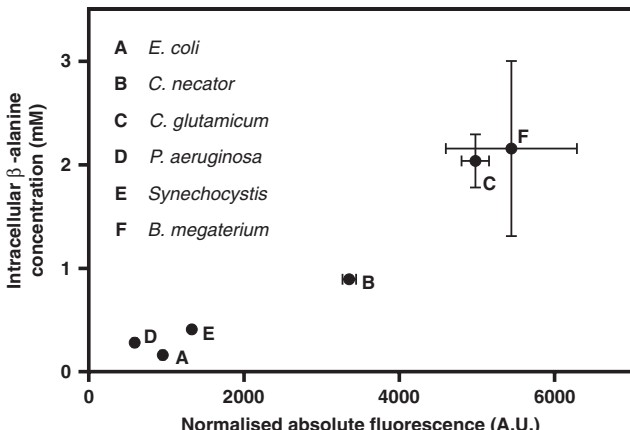

**Fig. 8 Correlation between biosensor output and intracellular β-alanine concentration.** Absolute normalised fluorescence values of *C. necator* cells, carrying plasmids containing the β-alanine biosensor and each a gene encoding a different homologue of L-aspartate 1-decarboxylase enzyme, correlated with their corresponding intracellular β-alanine concentration quantified by HPLC-UV. Cells were grown in MM. Samples were taken 6 h after supplementation with L-aspartate and L-arabinose. Data are mean ± SD, $n = 3$. Source data are provided as a Source Data file.

*C. necator* cells carrying these plasmids were grown in MM supplemented with both L-aspartate and L-arabinose to ensure an excess of substrate and to initiate the expression of the *panD* variants, respectively. The six selected homologues resulted in a ninefold range in fluorescence after 6 h (Fig. 8). To determine whether the biosensor output is indicative of product formation, metabolites were extracted from the cell pellets and the β-alanine concentrations were quantified using high-performance liquid

chromatography coupled with ultraviolet spectroscopy (HPLC-UV). Intracellular β-alanine titres correlated well with fluorescence levels (Fig. 8), highlighting the utility of this biosensor to aid in the selection of enzyme variants. Of the six tested homologues, the *Bacillus megaterium* PanD demonstrated the highest fluorescence and product titre. Moreover, the obtained results are in accordance with previous findings in *E. coli*, where PanD from *Corynebacterium glutamicum* has been shown to be significantly more active than the native *E. coli* PanD[46].

## Discussion
Thus far, inducible gene expression systems have been discovered primarily by research focussed on the experimental characterisation of individual metabolic pathways and their regulation. This traditional approach has allowed the identification of a substantial number of such systems, some of which have been developed into the widely used gene expression control devices, e.g., AraC/P$_{araBAD}$ and LacI/P$_{T7}$. However, this empirical approach, principally driven by interest in the pathway characterisation, usually delivers only a limited amount of original information on a specific inducible system. Recently, high-throughput applications, such as transcriptomics analysis, comparative genomics and promoter or metagenome library screens, have proved to be efficient methodologies and substantially enhanced the speed of discovery of effector-responsive promoters or even corresponding TR-promoter pairs[7–13]. However, even these strategies suffer from several limitations, since they either do not ensure identification of all essential components of an inducible system or are TR- or ligand-type specific.

In this study, we developed a methodical approach that allows inducible systems to be mined at the genome scale level enabling the extraction of information on all three components—the regulator, the inducible promoter and its corresponding effector. To

demonstrate the utility of our approach, the method was applied to the genome of the catabolically versatile *C. necator* H16. Sixteen putative inducible systems resulting from a pool that comprised over 400 TRs were identified in this single bacterium. With exception of the acetoin- and 3-HP-inducible systems[25,47], we identify and characterise 14 systems from *C. necator*. Two previously proposed regulators[48,49], responding to tartrate and sulfonatoacetate, were experimentally validated to be involved in transcription activation of their corresponding metabolic genes in this study. Furthermore, here, we report inducible systems (OapR/$P_{H16\_RS01330}$ and PhgR/$P_{H16\_RS05530}$) responding to β-alanine and phenylglyoxylate. Both compounds play an important role as building blocks in chemical synthesis or food biotechnology[31,50]. Among the characterised inducible systems, different types of TR were identified, including LysR, AsnC, MocR, IclR and MarR. Fifteen of these were activator or dual-function-type regulators, whereas the latter exhibited characteristics of a repressor (BadR/$P_{H16\_RS27200}$).

To further evaluate inducible systems, they were subjected to thorough quantitative characterisation assessing induction level, dynamics and homogeneity. Several inducible systems exceeded induction levels of the frequently utilised L-arabinose-inducible system[29], with the BenM/$P_{H16\_RS09790}$-inducible system achieving expression level of more than 11-fold higher than AraC/$P_{araBAD}$ in response to benzoate in *C. necator*. Four inducible systems responding to salicylate, sulfonatoacetate, tartrate and phenylglyoxylate exhibited a dynamic range of over 100-fold revealing that these systems are highly suitable to tightly control different levels of gene expression[51]. Along with a very high dynamic range of 650.7-fold, the NahR/$P_{H16\_RS08125}$-inducible system responds to nM concentrations of salicylate. This degree of sensitivity is equivalent to the most sensitive of characterised inducible systems, those based on anhydrotetracycline and cumate[52,53].

Orthogonal compatibility of inducible systems is a very important characteristic for designing multi-component and scalable circuits as well as sensory devices. These ideally require that functional genetic elements cross-react neither with the host genetic background nor between different heterologous systems. Twelve of the discovered inducible systems showed distinctive response only to their primary ligands. The capacity to independently drive the expression of multiple genes was exemplified by combining the salicylate- and 3,4-dihydroxybenzoate-inducible systems exquisitely demonstrating the potential of the characterised orthogonal switches for circuit design and other synthetic biology applications.

The utility of inducible systems was further demonstrated by applying individual promoter elements to control high levels of gene expression in *C. necator* (Fig. 2d) and by employing TR-promoter pairs in the model bacteria *E. coli* and *P. putida* (Fig. 3). Significantly, the adaptability of heterologous inducible system for gene expression control in other hosts was exemplified by engineering the β-alanine-inducible system for application in *E. coli* and *P. putida* (Fig. 4). In addition, the utility of the β-alanine biosensor was demonstrated by applying it to screen variants of L-aspartate 1-decarboxylase from different species and by identifying the *B. megaterium* homologue as most prominent to convert L-aspartate into β-alanine.

To conclude, the genome scale approach and inducible system evaluation pipeline presented in this paper, aids the discovery of metabolite-controlled systems. Further, it delivers quantitative data on inducible systems dynamics and orthogonality expanding the potential of developing tuneable regulatory circuits and biosensors. Furthermore, this generic approach can be utilised for mining inducible systems in any bacterial species, facilitating the expansion of the toolbox for synthetic biology and biotechnology applications.

## Methods

**Chemicals**. All chemicals employed as ligands in this study are listed in Supplementary Table 6.

**Base strains and media**. All strains used in this study are listed in Supplementary Table 7. *E. coli* TOP10 (Life Technologies) was used for cloning and plasmid propagation. For single time-point fluorescence measurements and flow cytometric analyses, bacterial strains were propagated in Luria-Bertani (LB) medium[54]. To determine the dose–responses and to evaluate inducer TR cross-reactivity, *C. necator* was cultivated in MM containing 1 g/L $NH_4Cl$, 9 g/L $Na_2HPO_4 \cdot 12H_2O$, 1.5 g/L $KH_2PO_4$, 0.2 g/L $MgSO_4 \cdot 7H_2O$, 0.02 g/L $CaCl_2 \cdot 2H_2O$ and 0.0012 g/L $(NH_4)5[Fe(C_6H_4O_7)_2]$ with 1 mL/L trace element solution SL7 (1.3 mL/L 25% (w/v) HCl, 0.07 g/L $ZnCl_2$, 0.1 g/L $MnCl_2 \cdot 4H_2O$, 0.062 g/L $H_3BO_3$, 0.190 g/L $CoCl_2 \cdot 6H_2O$, 0.017 g/L $CuCl_2 \cdot 2H_2O$, 0.024 g/L $NiCl_2 \cdot 6H_2O$ and 0.036 g/L $Na_2MoO_4 \cdot 2H_2O$) supplemented with 0.4% (w/v) sodium gluconate. If required, antibiotics were added to the growth medium at the following concentrations: 12.5 μg/mL tetracycline or 25 μg/mL chloramphenicol for *E. coli*, 50 μg/mL chloramphenicol for *C. necator* and 25 μg/mL tetracycline for *P. putida*. For solid media preparation, 15 g/L agar was added.

**Cloning and transformation**. Plasmid DNA was purified by using the QIAprep Spin Miniprep Kit (Qiagen). Microbial genomic DNA was extracted by employing the GenElute Bacterial Genomic DNA Kit (Sigma). DNA was amplified by PCR in 50 μL reactions using the Phusion High-Fidelity DNA polymerase from New England BioLabs (NEB). The Zymoclean Gel DNA Recovery Kit was employed to extract gel-purified linearised DNA. The NEBuilder Hifi DNA Assembly Master Mix, restriction enzymes and T4 DNA Ligase were purchased from NEB. All PCR-, digestion- and ligation reactions were set up according to the manufacturer's instructions.

For *E. coli* transformations, 50 μL of chemically competent *E. coli* TOP10 were mixed with 50 ng plasmid DNA, incubated on ice for 30 min, followed by a heat shock at 42 °C for 2 min and a subsequent incubation on ice for 2 min[54]. Cells were recovered in 450 μL of Super Optimal broth with Catabolite repression (SOC) medium (Invitrogen) at 37 °C for 1 h, plated on LB agar containing the appropriate antibiotic and incubated over night at 37 °C.

For *C. necator* transformations, 100 ng plasmid DNA were added to 100 μL of electrocompetent *C. necator* H16 in a pre-chilled electroporation cuvette (0.2 cm gap width, Bio-Rad) and incubated on ice for 5 min[55]. Electroporation was performed using a Bio-Rad Micropulser at 2.5 kV. Cells were recovered in 1 mL of SOC medium at 30 °C for 2 h, plated on LB agar containing the appropriate antibiotic and incubated at 30 °C for 2 days.

Electrocompetent *P. putida* KT2440 were freshly prepared from overnight cultures grown in LB medium at 30 °C and 200 rpm. A volume of 1 mL of cells were harvested by centrifugation at 16,000 × g for 5 min and washed with 1 mL of ice-cold 10% (v/v) glycerol. This step was repeated twice. Electroporation was conducted in an electroporation cuvette (0.2 cm gap width, Bio-Rad) using a Bio-Rad Micropulser at 12.5 kV[54]. Transformants were recovered in 1 mL of LB medium at 30 °C for 2 h, plated on LB agar containing the appropriate antibiotic and incubated over night at 30 °C.

**Plasmid construction**. Oligonucleotide primers were synthesised by Sigma-Aldrich (Supplementary Table 8). The *oapR* coding sequence was optimised for *E. coli* codon usage and synthesised by Life Technologies. The sequence can be found in the Supplementary Methods. Plasmids constructed by employing either the NEBuilder Hifi DNA assembly method or restriction enzyme-based cloning techniques[54] were validated by Sanger sequencing (Source BioScience, Nottingham, UK). A detailed assembly description for each plasmid is provided in the Supplementary Methods. Two versions of each plasmid containing an inducible system or an inducible promoter were constructed. One of the two versions contains a chloramphenicol resistance gene, the other one confers resistance to tetracycline. The former version was employed for evaluation of reporter gene expression in *C. necator* and *E. coli*, whereas the latter version was used in *P. putida*. To quantitatively evaluate the activity of various aspartate 1-decarboxylase (PanD) variants to convert L-aspartate into β-alanine, vectors were constructed that contain the original β-alanine biosensor and each one homologue of *panD* under control of the L-arabinose-inducible system. The *panD* genes were amplified from genomic DNA of the following species: *E. coli* MG1655, *C. necator* H16, *C. glutamicum* ATCC13032, *Pseudomonas aeruginosa* PAO1, *Synechocystis* sp. PCC6803 and *B. megaterium* DSM319. Key features of all plasmids used and generated in this study are summarised in Supplementary Table 9. The nucleotide sequences of the plasmids containing the 15 functional inducible systems pEH147, pEH134, pEH154, pEH155, pEH042, pEH148, pEH083, pEH157, pEH136, pEH137, pEH256, pEH158, pEH159, pEH161 and pEH052 have been deposited in the public version of the JBEI registry (https://public-registry.jbei.org) under the accession numbers JPUB_014465-JPUB_014479, respectively. The nucleotide sequence of plasmid pEH010 has been deposited under accession number JPUB_008754[47].

**Fluorescence measurements**. For quantification of RFP fluorescence at a single time point, individual colonies of freshly transformed bacterial cells were used to

inoculate 5 mL of LB containing the appropriate antibiotic in 50-mL conical centrifuge tubes. After incubation over night with orbital shaking at 200 rpm and 30 °C, *E. coli* and *P. putida* were diluted 1:50, and *C. necator* was diluted 1:20 into 5 mL of fresh LB medium containing the respective antibiotic. The exponentially growing cells with an $OD_{600}$ of 0.05–0.1 were supplemented with inducer to achieve a final concentration of 5 mM 4 h after the main culture had been set up or left uninduced. After a further incubation with orbital shaking at 30 °C and 200 rpm for 6 h, uninduced and induced cells with an $OD_{600}$ of 1.4-2.0 were pelleted by centrifugation at 16,000 × *g* for 4 min and resuspended in an equal volume of phosphate buffered saline (PBS). Subsequently, 100 μL of cells were transferred to a 96-well microtiter plate (flat and clear bottom, black; Greiner One International) and RFP fluorescence was quantified using an Infinite M1000 PRO microplate reader (Tecan). Fluorescence excitation and emission wavelengths were set to 585 nm and 620 nm, respectively. The gain factor was set manually to 100%. Absorbance was measured at 600 nm to normalise fluorescence by optical density. Prior normalisation, fluorescence and absorbance values were corrected by subtracting the auto fluorescence and absorbance of the culture medium.

To determine the dose–response of each individual inducible system, RFP fluorescence and absorbance were quantified over time. The precultures of *C. necator* cells were prepared as for the single time-point measurements in 2 mL of MM containing the appropriate antibiotic. After incubation over night, cells were diluted 1:20 into 5 mL of fresh MM containing the respective antibiotic and grown with orbital shaking at 30 °C and 200 rpm. At an $OD_{600}$ of 0.2, 142.5 μL of the exponentially growing cells were transferred to a 96-well microtiter plate. Each well was supplemented with 7.5 μL of stock inducer at the desired concentration. Fluorescence and absorbance were measured every 5 min over the time course of 6 h using the same excitation and emission wavelengths as for the single time-point measurements with the gain factor set to 80%. The fluorescence and absorbance values, recorded 80 min after the inducer had been added, were obtained from the time course data and used to calculate the absolute normalised fluorescence values corresponding to each inducer concentration.

**Production of β-alanine and metabolite extraction**. Real-time biosynthesis of β-alanine was monitored quantitatively by HPLC and by fluorescence output of *C. necator* harbouring both the β-alanine biosensor and different homologues of the gene encoding aspartate 1-decarboxylase (PanD). Single colonies of freshly transformed cells were used to inoculate 2 mL of MM in 50-mL conical centrifuge tubes. The preculture was incubated for 18 h with orbital shaking at 30 °C and 200 rpm. Subsequently, it was diluted 1:20 in 6 mL of fresh MM and incubated under the same conditions. At an $OD_{600}$ of 0.15–0.2, L-arabinose and L-aspartate were added to achieve the final concentrations of 1 mM and 20 mM, respectively. Samples of 0.6 mL were taken immediately, 2, 4 and 6 h after compound supplementation and used for absorbance and fluorescence measurements with the gain factor set to 80%. To quantify metabolites in the supernatant, cells were pelleted by centrifugation at 16,000 *g* for 5 min. A volume of 50 μL of the cell-free supernatant was kept at −80 °C until subjected to HPLC analysis.

To determine the intracellular concentration of β-alanine 6 h after supplementation with L-arabinose and L-aspartate, 1.5 mL of the remaining culture was pelleted by centrifugation at 16,000 *g* for 5 min in a preweighed 1.5-mL microcentrifuge tube. The supernatant was discarded and the same step repeated with another 1.5 mL of the culture. The cell pellet was resuspended in 1 mL of PBS and centrifuged as before. Subsequently, the supernatant was completely removed, the weight of the wet cell pellet determined using a fine balance and frozen over night at −80 °C. To extract metabolites, 50 μL of −40 °C methanol–water solution (60% v/v) was added to the pellet. The sample was mixed vigorously using vortex until completely resuspended. Cells were frozen at −80 °C for 30 min, thawed on ice and vortexed vigorously for 1 min. The freeze-thaw cycle was repeated three times. Subsequently, cells were pelleted by centrifugation at 26,000 *g* for 20 min at −10 °C. The supernatant was removed and kept at −80 °C. Cells were resuspended in another 50 μL of −40 °C methanol–water solution (60% v/v), subjected to three freeze-thaw cycles and centrifuged as before. The supernatant was pooled with the first collection and stored at −80 °C until used for HPLC analysis.

**Quantification of amino acids**. β-Alanine was quantified using a Dionex UltiMate 3000 HPLC system (Thermo Scientific) equipped with a Kinetex 5 μm EVO C18 100 Å LC 150 mm × 4.6 mm column (Phenomenex) and a photo diode array (UV-VIS) detector measuring the absorption at 338 nm and 210 nm. The sample was prepared by adding 950 μL of 50% methanol diluent (HPLC-grade) to 50 μL of either the cell-free supernatant or the extract containing the intracellular metabolites. After the sample was mixed by vortexing, it was filtered into a HPLC vial using a 0.2 μm syringe filter. Derivatisation of amino acids with fluoraldehyde *o*-phthalaldehyde (OPA) was performed automatically by the autosampler before injection. The autosampler programming instructions can be found in Supplementary Table 10. Preparation of OPA reagent was adapted from Roth[56] with slight modifications: the reagent is composed of OPA (0.8 g/L) diluted in HPLC-grade methanol (10 mL/L), added to a 0.1 M $KH_2PO_4$ buffer at pH 10.4 supplemented with β-mercaptoethanol (2 mL/L). The HPLC method was adapted from Phenomenex HPLC application ID 23092 (https://phenomenex.com/Application/Detail/23092). Briefly, two mobile phases were used: mobile phase A was 20 mM $KH_2PO_4$ adjusted to pH 7.2 with KOH, while mobile phase B was methanol/

acetonitrile (50/50 v/v, HPLC-grade). All samples and reagents were kept at 4 °C throughout the analysis. The column was operated at 30 °C. The injection volume was 5 μL and samples were run for 23 min. The separation was achieved with a flow rate of 1 mL/min and followed the gradient programme in Supplementary Table 11. Data analysis was performed using Chromeleon 7 (Thermo Scientific). Metabolite concentrations were quantified using calibration curves generated from running standards of known concentrations which were prepared the same as the samples.

**Calculation of intracellular β-alanine concentration**. The volume of the cell pellet ($V_{pellet}$) was calculated by dividing the weight of the wet cell pellet by the cell density of 1.105 g/mL[57]. The cell density of *C. necator* was denoted to be the same as of *E. coli*, assuming that under the conditions tested no polyhydroxybutyrate (PHB) had been accumulated. Together with the volume of extraction solvent added to the pellet ($V_{solvent}$), $V_{pellet}$ was used to calculate the intracellular molar concentration of β-alanine using formula (1):

$$C_{intracellular} = \left( \frac{V_{pellet} + V_{solvent}}{V_{pellet}} \right) \times C_{extract} \tag{1}$$

The remaining parameters correspond to the intracellular molar concentration of β-alanine ($C_{intracellular}$) and the molar concentration of β-alanine in the extract ($C_{extract}$) determined by HPLC-UV analysis.

**Mathematical modelling**. To obtain system parameters that can be used for synthetic circuit design, absolute normalised fluorescence values (RFP) were plotted as a function of inducer concentration using software GraphPad Prism 7. Subsequently, a non-linear least-squares fit was performed using the Hill function (2):

$$RFP(I) = b_{max} \times \frac{I^h}{K_m^h + I^h} + b_{min} \tag{2}$$

The parameters correspond to the maximum level of reporter output ($b_{max}$), the concentration of inducer ($I$), the Hill coefficient ($h$), the inducer concentration that mediates half-maximal reporter output ($K_m$) and the basal level of fluorescence output ($b_{min}$). Relative normalised fluorescence values as shown in Fig. 5a were obtained by dividing absolute normalised fluorescence values at a specific inducer concentration after subtraction of the absolute normalised fluorescence of the uninduced cells by the corresponding maximum level of fluorescence output $b_{max}$.

The dynamic range $\mu$ was calculated with formula (3):

$$\mu = \frac{b_{max}}{b_{min}} \tag{3}$$

The corresponding standard deviation $\sigma_\mu$ was calculated using Eq. (4):

$$\sigma_\mu = \mu \times \sqrt{\left( \frac{\sigma_{b_{max}}}{b_{max}} \right)^2 + \left( \frac{\sigma_{b_{min}}}{b_{min}} \right)^2} \tag{4}$$

The standard deviation of the maximum level of reporter output was obtained from Prism, whereas the standard deviation of the basal level of fluorescence output was calculated from the absolute normalised fluorescence values of the uninduced cells.

**Flow cytometry**. For evaluation of induction homogeneity and cross-reactivity by flow cytometry, single colonies of freshly transformed *C. necator* cells were used to inoculate 5 mL of LB containing 50 μg/mL chloramphenicol in 50-mL conical centrifuge tubes. After incubation over night with orbital shaking at 30 °C and 200 rpm, cells were diluted 1:50 in 5 mL of fresh LB medium containing the antibiotic. Inducers were added to the logarithmically growing cells 4 h after further incubation. Samples were taken 2 h after supplementation with inducer. Cells were pelleted by centrifugation at 4,000 *g* for 5 min and resuspended in cold and sterile filtered PBS to an $OD_{600}$ of 0.01. The cells were kept on ice until analysed using an Astrios EQ flow cytometer (Beckman Coulter). RFP fluorescence was measured with a 561 nm laser and a 614/20 nm emission band-pass filter. GFP fluorescence was quantified with a 488 nm laser and a 516/28 nm emission band-pass filter. The voltage of the photomultiplier tube was set to 400 V and the area and height gain was adjusted to 1.0. At least 100,000 events were collected for each sample. Data analysis was performed using software Kaluza 1.5 (Beckman Coulter).

**Cross-reactivity screen**. The activity of effectors against non-cognate promoters was evaluated using an integrated robotic platform (Beckman Coulter). The *C. necator* preculture was set up by inoculating 2 mL of chloramphenicol-containing MM with a single colony of freshly transformed bacterial cells. After incubation for 18 h with orbital shaking at 30 °C and 200 rpm, the bacterial culture was diluted 1:50 in 50 mL of fresh MM containing the antibiotic in 250-mL baffled shake flasks. The cells were grown for another 4 h under the same conditions until an $OD_{600}$ of 0.15–0.2 was reached. After pouring the bacterial culture into a 250-mL reservoir (Thermo Fisher), 142.5 μL were dispensed into a black 96-well microtiter plate (the same that was used for the fluorescence measurements) using a liquid handling robotic platform (Biomek FXp, Beckman Coulter). The workflow generated using

software SAMI EX (Beckman Coulter) is illustrated in Supplementary Fig. 10. Inducers were dissolved to a final concentration of 100 mM (except in case of L-2-hydroxyglutarate which was dissolved to a final concentration of 10 mM) and transferred to a 96-deep-well plate (2.0 mL square wells with round bottoms, STARLAB International GmbH). Using a Biomek FXp, 7.5 μL of stock inducer were added to the *C. necator* culture in the microtiter plate. Fluorescence and absorbance measurements were taken immediately, 6, 12 and 18 h after supplementation with inducer by an integrated SpectraMax 3i plate reader (Molecular Devices). The same plate reader settings were used as for the measurements taken using the Infinite M1000 PRO microplate reader. In between the measurements, the plates were kept in an integrated Cytomat2 shaking incubator (Thermo Fisher Scientific) at 30 °C and 600 rpm. The workflow generated using software SAMI EX (Beckman Coulter) is illustrated in Supplementary Fig. 11.

The relative induction (in %) is calculated using Eq. (5):

$$Relative\ induction\ (\%) = 100 \times \left( \frac{FL_{compound} - FL_{uninduced}}{FL_{primary\ inducer} - FL_{uninduced}} \right) \quad (5)$$

*FL* corresponds to the OD-normalised absolute fluorescence values.

**Reporting summary**. Further information on research design is available in the Nature Research Reporting Summary linked to this article.

## Data availability

All data generated or analysed during this study are included in this published article (and its Supplementary Information files). The nucleotide sequences of the plasmids containing the 16 functional inducible systems have been deposited in the public version of the JBEI registry (https://public-registry.jbei.org) under the accession numbers JPUB_014465-JPUB_014479 and JPUB_008754. The nucleotide sequences of the plasmids containing the 15 functional inducible systems pEH042, pEH052, pEH083, pEH134, pEH136, pEH137, pEH147, pEH148, pEH154, pEH155, pEH157, pEH158, pEH159, pEH161 and pEH256 have been deposited NCBI GenBank under the accession numbers MT024789, MT024790, MT024791, MT024792, MT024793, MT024794, MT024795, MT024796, MT024797, MT024798, MT024799, MT024800, MT024801, MT024802 and MT024803, respectively. The source data underlying Fig. 2c, d, 3a–d, 4b, c, 5a, 6a, 7, 8, Table 1, Supplementary Figs. 3b, 4, 5, and Supplementary Tables 4 and 5 are provided as a Source Data file or available from the corresponding authors upon reasonable request.

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

## Acknowledgements

This work was supported by the Biotechnology and Biological Sciences Research Council (BBSRC; grant number BB/L013940/1) and the Engineering and Physical Sciences Research Council (EPSRC) under the same grant number. We thank The University of Nottingham for providing SBRC-DTPProg Ph.D. studentship to E.K.R.H. and Ph.D. studentship to A.C.P., Stephan Heeb for gifting strain *P. putida* KT2440, Toby Pendlebury and Simona Lukosiute for assistance with plasmid assembly, David Onion for assistance with flow cytometry, R. Biedendieck, S. Grigoriou and S. Craig for gifting microbial genomic DNA, and all members of the SBRC who helped to carry out this research.

## Author contributions

E.K.R.H. and N.M. conceived the project, designed and performed the experiments, analysed the results and wrote the manuscript. A.C.P. conducted fluorescence assays in *P. putida*. M.J. assisted with the robotic platform for cross-reactivity screen. M.A. designed the HPLC method and assisted with HPLC analysis. N.P.M. analysed the results and wrote the manuscript. All authors read and approved the final manuscript.

## Competing interests

The authors declare no competing interests.
