## [Peer Review File · Nature Communications]

Reviewers' Comments:

Reviewer #1:

Remarks to the Author:

The authors present an interesting approach to find responsive transcription factors and hence biosensors through a bioinformatics pipeline. This work is highly interesting for synthetic biology research as it creates new inducible systems. Overall, the paper is quite scientifically sound and should be published upon some minor edits.

The authors chose to conduct this work with the use of *Cupriavidus necator*. This is not a conventional organism and hence the decision/choice to use this organism was a bit unclear in the text. What was the rationale? Would results have been the same in other types of bacteria? What is special about this organism (does this have more LysR type systems than other bacteria?)

Table 1 was a bit hard to follow as it sat in the text without a full legend and definition of the column headings. It was hard to know the definitions for dynamic range (without seeing this fully in the mathematical formulas section at the end of the paper). I am still unsure what "h" means in the last column.

The full sequences of the regulators and promoters should be provided in the supplementary information so that others can replicate these constructs. This was done for DNA sequence for OapR, but not the remaining elements or promoters.

Reviewer #2:

Remarks to the Author:

The authors of this manuscript followed a genome-wide approach for the identification of new inducible expression systems in bacteria, which can be used for the construction of novel transcription factor-based biosensors. For this purpose, they mined the genome of the chemolithoautotrophic bacterium *Cupriavidus necator*. They identified and further characterized 16 inducible systems and tested the orthogonality of these systems by transferring them to the biotechnological model species *Pseudomonas putida* and *E. coli*. Out of the 16 systems characterized, the authors identified two systems responding to β -alanine and phenylglyoxylate for which no TF-based biosensors have been described so far. The authors conducted an intensive characterization of the systems with and without TF-encoding gene, tested induction dynamics and heterogeneity and potential cross-reactivity of the systems. The authors performed proof-of-concept studies for selected systems to emphasize their systems for synthetic circuit design. Overall, the authors performed an intensive effort to identify novel inducible systems in *C. necator* for application in biotechnology or synthetic biology and I believe that the results may be useful for different applications. My main criticism actually is, however, that the study does not significantly add to the status quo in the field. Furthermore, developed biosensors were not applied for metabolic engineering/screening of production strains or enzyme screenings.

- 1) The principle that potential TF/promoter pairs are selected on the genomic organization has been frequently and successfully applied by several studies.
- 2) The study aims to improve the diversity of biosensors by developing a 'methodical pipeline' (for me this is rather an approach than a method...). However, this pipeline only covers a specific type of regulatory circuits, where a TF is divergently expressed of a gene cluster encoding catalysts, and its use only results in the discovery of 16 sensor-promoter pairs, of which only 2 ligands did not have a sensor described previously. In these two cases, the authors fall short in communicating why these substances are especially relevant or interesting for synthetic biology and/or biotechnology.
- 3) The overall characterization approach is a bit shallow. By no means are the effects of ligands on growth, the uptake of ligands and the degradation of ligands discussed – these could have a major

effect on sensor response, especially in combination with 6h of incubation time. There is not a single graph on fluorescence and/or growth over time on 37 pages of supplemental material.

4) Figure 5: Here the extracellular effector molecule concentration is correlated to the fluorescence output. This may NOT necessarily show a linear correlation to the intracellular concentration and should be more emphasized here. Furthermore, the presentation in this Figure is from my point of view not correct. The authors set the maximum level of reporter output to 91% - WHY? The cells may be simply limited in terms of uptake/toxicity(?) of this particular effector. This does not mean that the system is anyhow close to saturation. In fact, many measurements shown in Fig. 5A do not reach saturation (e.g. see TtdR), but the fit implies otherwise.

5) The M&M should be more elaborate. For example it is not described at which biomass concentrations the sensor responses were measured and how the controls were measured (was the incubation time, biomass concentration and growth phase similar?).

Further comments

- The storyline could be better – it is now not completely clear what the main point of this study is.
- Figure 5. The FACS plots (B) should not be in 2d. 3d doesn't add any information here.
- It is not clear why all experiments were performed on complex media, minimal media could give more reproducible results.
- I would suggest to provide an overview on the genomic organization of all chosen systems mined from the *C. necator* genome
- Figure legends: Please provide relevant information. For example in figure 2: which organisms, plasmids or genomically integrated constructs, which reporter is used?
- Figure 3: Give effectors in the Figure – this enhances readability
- Page 12, lane 21-22: Why are these two molecules in particular interesting for synthetic biology and biotechnological applications?

We thank the reviewers for their careful attention to our work which has led to a revised version of the manuscript, now much improved due to their comments. Below we respond to their questions; our answers are in blue font.

Reviewers' comments:

Reviewer #1 (Remarks to the Author):

The authors present an interesting approach to find responsive transcription factors and hence biosensors through a bioinformatics pipeline. This work is highly interesting for synthetic biology research as it creates new inducible systems. Overall, the paper is quite scientifically sound and should be published upon some minor edits.

The authors chose to conduct this work with the use of *Cupriavidus necator*. This is not a conventional organism and hence the decision/choice to use this organism was a bit unclear in the text. What was the rationale? Would results have been the same in other types of bacteria? What is special about this organism (does this have more LysR type systems than other bacteria?)

We thank reviewer for his/her comments. We would like to highlight that this is a first report on the application of a generic genome-wide approach to identify transcription factor-based metabolite-inducible systems. This approach can be applied to any bacterial species and its utility is exemplified by application to the metabolically versatile bacterial species such as *C. necator*. In the revised version of manuscript we point out that 'we address the deficiencies associated with the identification of metabolite-responsive inducible systems by interconnecting information on ligand metabolism, TR genes and gene clusters responsible for the catabolism of the corresponding ligand. A generalised genome-wide approach is established to discover new native systems independent of '... bacterial species utilised as a genetic resource'. To explain the use of this particular organism, we highlight that 'the approach ... was applied in the chemolithoautotrophic bacterium *Cupriavidus necator* H16, known for its metabolic versatility and diverse gene expression regulation'.

Finally, we would like to note that amongst transcription factor-based inducible systems identified in this study, there were other types of regulators, including AsnC, MocR, IclR, and others, along the LysR-type regulators (Supplementary Table 1).

Table 1 was a bit hard to follow as it sat in the text without a full legend and definition of the column headings. It was hard to know the definitions for dynamic range (without seeing this fully in the mathematical formulas section at the end of the paper). I am still unsure what "h" means in the last column.

Thank you for this observation and pointing out the missing information about definitions in Table 1. In the revised version of the manuscript we provide additional information under Table 1 as follows: 'dynamic range is defined as the -fold increase in fluorescence calculated by dividing the maximum level of fluorescence output by the basal level of fluorescence output; bK_m represents the inducer concentration at which the half-maximal activation of the inducible system is achieved; $^c h$ – Hill coefficient.' Table 1 has been amended accordingly.

The full sequences of the regulators and promoters should be provided in the supplementary information so that others can replicate these constructs. This was done for DNA sequence for OapR, but not the remaining elements or promoters.

Thank you for this recommendation. We deposited the full sequences of the regulators and promoters for each inducible system in the public version of the JBEI registry. The sequence data will become publicly available immediately after publication. The revised version of the manuscript reads as follows: 'The nucleotide sequences of the plasmids containing the 15 functional inducible systems pEH147, pEH134, pEH154, pEH155, pEH042, pEH148, pEH083, pEH157, pEH136, pEH137, pEH256, pEH158, pEH159, pEH161, and pEH052 have been deposited in the public version of the JBEI registry (<https://public-registry.jbei.org>) under the accession numbers JPUB_014465-JPUB_014479, respectively. The nucleotide sequence of plasmid pEH010 has been deposited previously⁴⁹ under accession number JPUB_008754.'

Reviewer #2 (Remarks to the Author):

The authors of this manuscript followed a genome-wide approach for the identification of novel inducible expression systems in bacteria, which can be used for the construction of novel transcription factor-based biosensors. For this purpose, they mined the genome of the chemolithoautotrophic bacterium *Cupriavidus necator*. They identified and further characterized 16 inducible systems and tested the orthogonality of these systems by transferring them to the biotechnological model species *Pseudomonas putida* and *E. coli*. Out of the 16 systems characterized, the authors identified two systems responding to β -alanine and phenylglyoxylate for which no TF-based biosensors have been described so far. The authors conducted an intensive characterization of the systems with and without TF-encoding gene, tested induction dynamics and heterogeneity and potential cross-reactivity of the systems. The authors performed proof-of-concept studies for selected systems to emphasize their systems for synthetic circuit design.

Overall, the authors performed an intensive effort to identify novel inducible systems in *C. necator* for application in biotechnology or synthetic biology and I believe that the results may be useful for different applications. My main criticism actually is, however, that the study does not significantly add to the status quo in the field. Furthermore, developed biosensors were not applied for metabolic engineering/screening of production strains or enzyme screenings.

We thank reviewer for his/her comments. We would like to highlight that this is a first report on the application of a generic genome-wide approach to identify transcription factor-based metabolite-inducible systems. This approach can be applied to any bacterial species and its utility is exemplified by application to the metabolically versatile bacterial species, such as *C. necator*. Furthermore, in the revised version of the manuscript we added a section providing data (Figure 8) on application of the β -alanine-biosensor for screening of enzyme homologs with aspartate decarboxylase activity. In total we analysed aspartate decarboxylases from six different microorganisms to identify the ones that efficiently convert L-aspartate into β -alanine. We were able to demonstrate that intracellular β -alanine concentrations quantified using HPLC-UV analysis correlate well with the biosensor output. This additional experiment highlights the potential of this newly identified inducible system to be applied in synthetic biology and biotechnology applications.

1) The principle that potential TF/promoter pairs are selected on the genomic organization has been frequently and successfully applied by several studies.

We agree with reviewer that the principle of TF/promoter pairs selected on the genomic organization has been applied previously which we extensively cite in our paper introduction (references 12 to 18). We would like to point out that in our proposed approach we extend this principle by incorporating metabolic relevance of the operon that is controlled by TF/promoter pairs,

which delivers much more robust prediction and results in very high success rate for metabolite/TF/promoter triplet identification. In this study we achieve more than 90% success rate by applying our proposed approach.

2) The study aims to improve the diversity of biosensors by developing a 'methodical pipeline' (for me this is rather an approach than a method...). However, this pipeline only covers a specific type of regulatory circuits, where a TF is divergently expressed of a gene cluster encoding catalysts, and its use only results in the discovery of 16 sensor-promoter pairs, of which only 2 ligands did not have a sensor described previously. In these two cases, the authors fall short in communicating why these substances are especially relevant or interesting for synthetic biology and/or biotechnology.

We would like to point out that in addition to the 2 novel systems, 12 of the other 14 systems from *C. necator* have never been cloned, let alone characterised. To our knowledge, only six of them (mined from other organisms) have been previously evaluated for their ability to control gene expression in other microbes (see Supplementary Table 7). The Discussion of revised manuscript extra text was added and this part of the manuscript now includes following text: ' . Both compounds play an important role as building blocks in chemical synthesis or food biotechnology. In addition, 14 of the 16 inducible systems from *C. necator* identified in this study have never been characterised previously. Furthermore, tartrate- and sulfonatoacetate-inducible systems, which were proposed previously, ... were experimentally validated.'

We think that this study not only promotes their application as biosensors but also provides parameters required for synthetic circuit design. The manuscript text and title were revised to emphasise that a big part of this study aimed at characterising the identified systems. As proposed by reviewer, we have changed term 'methodical pipeline' to 'approach'. Furthermore, we included section about application of β -alanine as important precursor which reads as follows: ' β -Alanine is an intermediate compound for the synthesis of industrially relevant nitrogen-containing platform chemicals, including acrylamide, acrylonitrile, and poly- β -alanine (also known as nylon-3)^{30, 31}. Furthermore, it is a precursor of the dipeptides carnosine and anserine which have been demonstrated to improve cognitive functions and physical capacities in humans^{32, 33}'. Finally, we note in the Discussion that both compounds (β -alanine and phenylglyoxylate) play an important role as building blocks in chemical synthesis or food biotechnology.

3) The overall characterization approach is a bit shallow. By no means are the effects of ligands on growth, the uptake of ligands and the degradation of ligands discussed – these could have a major effect on sensor response, especially in combination with 6h of incubation time. There is not a single graph on fluorescence and/or growth over time on 37 pages of supplemental material.

To address the reviewers comment, in the revised version of the manuscript we provide data (Supplementary Fig. 4 and 5) on growth and fluorescence profile over time for all 16 inducible systems characterised in the paper. Moreover we added an additional paragraph covering the kinetics of induction and the effects of the ligands on cell growth in the Results section 'Parameterisation of inducible systems'. In the revised version of manuscript, the relevant text reads as following: 'Most of the compounds had a beneficial effect on growth and no toxicity was observed for any effector at the tested concentration of 5 mM (Supplementary Fig. 5). L-tyrosine and 3,4-dihydroxybenzoate had the most significant impact on growth resulting in more than a 2-fold increase in cell density most likely due to ligand catabolism. Therefore, the effector consumption plays an important role in the kinetics of induction. In order to parameterise the identified systems, assumptions must be implemented that account for these factors, including ligand uptake and metabolism.'

We furthermore point out that 'in order to parameterise the identified systems, assumptions must be implemented that account for ... ligand uptake and metabolism'. Consequently, we selected a time-point for the generation of the dose-response curve at which ligand degradation is negligible. As a result of the mathematical modelling, K_m values for the GABA-, tartrate-, and sulfonatoacetate-inducible systems were obtained that are too high to be biologically meaningful. In the revised version of the manuscript we point out that, based on the results of the response-fitting, 'for the other 13 systems ligand uptake is assumed not to be limiting'.

4) Figure 5: Here the extracellular effector molecule concentration is correlated to the fluorescence output. This may NOT necessarily show a linear correlation to the intracellular concentration and should be more emphasized here. Furthermore, the presentation in this Figure is from my point of view not correct. The authors set the maximum level of reporter output to 91% - WHY? The cells may be simply limited in terms of uptake/toxicity(?) of this particular effector. This does not mean that the system is anyhow close to saturation. In fact, many measurements shown in Fig. 5A do not reach saturation (e.g. see TtdR), but the fit implies otherwise.

We thank reviewer for this comment. The revised version of the manuscript reads as follows: 'In contrast to most of the inducible systems that operate in the μM -range, GabR, TtdR, and SauR seem to respond to effector concentrations 3 to 5 orders of magnitude higher than NahR... It should be noted that the extracellular effector concentration may not necessarily correlate with the ligand concentration inside the cell ultimately dictating the level of gene expression. Ligand uptake limitations may therefore result in inaccurate parameters as it might be the case for the GABA-, tartrate-, and sulfonatoacetate-inducible systems.'

In Figure 5A, we chose to set the maximum level of reporter output, calculated using the Hill function, to 91%. We followed a previous study that performed data normalisation by dividing each data point in the graph by 110% ($100/110=91\%$) of the highest value for visualisation purposes. However, we agree with reviewer that this may seem arbitrary. We revised Fig. 5. The maximum level of reporter output b_{max} was now set to 100%.

We also agree with reviewer that the tartrate-, sulfonatoacetate-, and GABA-inducible systems (TtdR, SauR, and GabR, respectively) do not reach saturation. In these three cases we point out in the revised version of the manuscript that this may be due to ligand uptake limitation (page Y-Z, line X). However, for all systems (except in case of GabR) we were able to measure the reporter response at inducer concentrations above K_m . From a mathematical point of view, K_m represents the inflection point. In order to accurately fit the data points to the model, the fluorescence output at K_m is required. In case of the GABA-inducible system we were not able to supplement the growth medium with high enough concentrations of GABA to determine the reporter output close to K_m , hence the model did not yield a meaningful b_{max} . For all other systems, we made sure to obtain data points at inducer concentrations below and above K_m (and especially close to K_m) in order to accurately apply the model.

5) The M&M should be more elaborate. For example it is not described at which biomass concentrations the sensor responses were measured and how the controls were measured (was the incubation time, biomass concentration and growth phase similar?).

To address reviewers comment, in the revised version of the manuscript we provide the biomass concentrations at which the inducers were added and when the sensor responses were measured. Uninduced sample controls were treated as induced samples. The revised version of the manuscript reads as follows: 'The exponentially growing cells with an OD_{600} of 0.05-0.1 were supplemented with inducer to achieve a final concentration of 5 mM 4 h after the main culture had been set up or left

uninduced. After a further incubation with orbital shaking at 30 °C and 200 rpm for 6 h, uninduced and induced cells with an OD₆₀₀ of 1.4-2.0 were pelleted by centrifugation at 16,000g for 4 min and resuspended in an equal volume of phosphate buffered saline (PBS).'

Further comments

- The storyline could be better – it is now not completely clear what the main point of this study is.

Thank you for this comment. In the revised version of the manuscript we exemplify main point of this study as the genome-wide approach for identification, validation and characterisation of native inducible systems, which can be applied to any microorganism and is exemplified by applying it the biotechnologically relevant and metabolically versatile bacterium *C. necator*. In the revised introduction of the manuscript, 'we address the deficiencies associated with the identification of metabolite-responsive inducible systems by pairing interconnecting information on ligand metabolism, TR genes with and gene clusters responsible for the catabolism of the corresponding ligand and establishing a generalised workflow genome-wide approach to discover new native systems independent of their belonging to a specific family of regulators, or the class of compounds they respond to or bacterial species utilized as a genetic resource. Newly discovered systems are validated for response to proposed compounds and their comprehensive characterisation is performed'.

- Figure 5. The FACS plots (B) should not be in 2d. 3d doesn't add any information here.

We agree with reviewer that 3D does not necessarily add much information, however, this type of illustration makes it easier to distinguish between the individual peaks.

- It is not clear why all experiments were performed on complex media, minimal media could give more reproducible results.

The single time-point measurements were performed in complex media to rule out any ligand-independent effects the species-specific minimal media may have on system activation in the three selected microorganisms. The system parameterisation, the orthogonality screen, the library screen and the biosensor-assisted screening of enzyme variants in *C. necator* was performed in the minimal medium.

- I would suggest to provide an overview on the genomic organization of all chosen systems mined from the *C. necator* genome

We thank reviewer for this remark. The genomic organisation of all identified systems can now be found in Supplementary Fig. 1.

- Figure legends: Please provide relevant information. For example in figure 2: which organisms, plasmids or genomically integrated constructs, which reporter is used?

Figure 2 presents single-time point measurements performed in *C. necator*. We added plasmid identifiers to the figure and highlighted the use of the RFP reporter in the figure legend as suggested by the reviewer.

Figure 3: Give effectors in the Figure – this enhances readability

Thank you for this suggestion. We modified Figure 3 to contain effector names and plasmid identifiers to improve clarity.

Page 12, lane 21-22: Why are these two molecules in particular interesting for synthetic biology and biotechnological applications?

We thank reviewer for this comment. In the revised version of the manuscript we highlight the importance of the selected compound for synthetic and biotechnology applications.

Reviewers' Comments:

Reviewer #2:

Remarks to the Author:

The authors have satisfactorily addressed the Points I have raised in my previous Review. Especially the Screening of PanD variants makes a much more convincing Point of the novel biosensors.

- I have just one more question with respect to the minimal medium. Is this really the one described by Schlegel et al in 1961 or is a more recent variant used?

2nd revision

We thank the reviewers for their careful attention to our work which has led to a revised version of the manuscript, now much improved due to their comments. Below we respond to their questions; our answers are in blue font.

Reviewers' comments:

Reviewer #2 (Remarks to the Author):

The authors have satisfactorily addressed the Points I have raised in my previous Review. Especially the Screening of PanD variants makes a much more convincing Point of the novel biosensors.

- I have just one more question with respect to the minimal medium. Is this really the one described by Schlegel et al in 1961 or is a more recent variant used?

We thank reviewer for his/her question. To address this concern and make information more easily accessible we provide complete composition of the minimal medium and remove reference Schlegel et al 1961. In the revised version of manuscript the following text is added to Methods/Base strains and media section: minimal medium (MM) containing 1 g/L NH₄Cl, 9 g/L Na₂HPO₄·12H₂O, 1.5 g/L KH₂PO₄, 0.2 g/L MgSO₄·7H₂O, 0.02 g/L CaCl₂·2H₂O, 0.0012 g/L (NH₄)₅[Fe(C₆H₄O₇)₂] with 1 mL/L trace element solution SL7 (1.3 mL/L 25% (w/v) HCl, 0.07 g/L ZnCl₂, 0.1 g/L MnCl₂·4H₂O, 0.062 g/L H₃BO₃, 0.190 g/L CoCl₂·6H₂O, 0.017 g/L CuCl₂·2H₂O, 0.024 g/L NiCl₂·6H₂O, 0.036 g/L Na₂MoO₄·2H₂O) supplemented with 0.4% (w/v) sodium gluconate.